# Time-resolved parameterization of aperiodic and periodic brain activity

**Luc Edward Wilson, Jason da Silva Castanheira, Sylvain Baillet***

McConnell Brain Imaging Centre, Montreal Neurological Institute, McGill University, Montreal, Canada

**Abstract** Macroscopic neural dynamics comprise both aperiodic and periodic signal components. Recent advances in parameterizing neural power spectra offer practical tools for evaluating these features separately. Although neural signals vary dynamically and express non-stationarity in relation to ongoing behaviour and perception, current methods yield static spectral decompositions. Here, we introduce Spectral Parameterization Resolved in Time (SPRiNT) as a novel method for decomposing complex neural dynamics into periodic and aperiodic spectral elements in a time-resolved manner. First, we demonstrate, with naturalistic synthetic data, SPRiNT's capacity to reliably recover time-varying spectral features. We emphasize SPRiNT's specific strengths compared to other time-frequency parameterization approaches based on wavelets. Second, we use SPRiNT to illustrate how aperiodic spectral features fluctuate across time in empirical resting-state EEG data (n=178) and relate the observed changes in aperiodic parameters over time to participants' demographics and behaviour. Lastly, we use SPRiNT to demonstrate how aperiodic dynamics relate to movement behaviour in intracranial recordings in rodents. We foresee SPRiNT responding to growing neuroscientific interests in the parameterization of time-varying neural power spectra and advancing the quantitation of complex neural dynamics at the natural time scales of behaviour.

## Editor's evaluation

The paper addresses the highly timely question of how to quantify aperiodic and periodic neural activity. This was done by extending previous work by embracing time-resolved parametrization of both simulated, noninvasive EEG and intracranial data. The new approach is termed Spectral Parametrization Resolved in Time (SPRiNT) and the paper shows that the slope of aperiodic activity is linked with both behavior and age. The method thus demonstrates the importance of evaluating the state-dependence of aperiodic activity and dynamic properties of oscillatory components in a time-resolved manner, and we believe that this approach would be of great interest to researchers analyzing human electrophysiological data to address clinical and cognitive neuroscience questions.

*For correspondence:
sylvain.baillet@mcgill.ca

**Competing interest:** The authors declare that no competing interests exist.

## Introduction

The brain constantly expresses a repertoire of complex dynamics related to behaviour in health and disease. Neural oscillations, for instance, are rhythmic (periodic) components of brain activity that emerge from a background of arrhythmic (aperiodic) fluctuations recorded with a range of electrophysiological techniques at the mesoscopic scale (*Buzsáki, 2006*). Brain oscillations and their rhythmic dynamics have been causally linked to individual behaviour and cognition (*Albouy et al., 2017*) and shape brain responses to sensory stimulation (*Samaha et al., 2020*).

Current methods for measuring the time-varying properties of neural fluctuations include several time-frequency decomposition techniques such as Hilbert, wavelet, and short-time Fourier signal transforms (*Bruns, 2004*; *Cohen, 2014*), and more recently, empirical mode decompositions (*Huang*

*et al., 1998*) and time-delay embedded hidden Markov models (*Quinn et al., 2018*). Following spectral decomposition, rhythmic activity within the empirical bands of electrophysiology manifests as peaks of signal power (*Buzsáki and Watson, 2012*; *Cohen, 2014*). However, time-resolved signal power decompositions (spectrograms) do not explicitly account for the presence of aperiodic signal components, which challenge both the detection and the interpretability of spectral peaks as genuine periodic signal elements (*Donoghue et al., 2020*). This is critical as aperiodic and periodic components of neural activity represent distinct, although possibly interdependent physiological mechanisms (*Gao et al., 2017*).

Aperiodic neural activity is characterized by a reciprocal distribution of power with frequency (1 /f), which can be parameterized with two scalars: exponent and offset. These parameters are physiologically meaningful: current constructs consider the offset as reflecting neuronal population spiking and the exponent as related to the integration of synaptic currents (*Voytek and Knight, 2015*) and reflecting the balance between excitatory (E) and inhibitory (I) currents (i.e., the larger the exponent, the stronger the inhibition; *Chini et al., 2021*; *Gao et al., 2017*; *Waschke et al., 2021*). Aperiodic neural activity is ubiquitous throughout the brain (*He, 2014*), and it differentiates healthy ageing (*Cellier et al., 2021*; *Donoghue et al., 2020*; *Hill et al., 2022*; *Ostlund et al., 2022*; *Schaworonkow and Voytek, 2021*; *Voytek et al., 2015*) and is investigated as a potential marker of neuropsychiatric conditions (*Molina et al., 2020*) and epilepsy (*van Heumen et al., 2021*). Though the study of aperiodic neural activity has recently advanced, unanswered questions remain about its functional relevance, which requires an expanded toolkit to track their evolution across time and the broadest possible expressions of behaviour.

One little studied aspect of aperiodic activity is its fluctuations, both spontaneously over time, and in association with task and mental states. Baseline normalization is a common approach to compensate for aperiodic contributions to spectrograms (*Cohen, 2014*), with the underlying assumption, however, that characteristics of aperiodic activity (exponent and offset) remain unchanged throughout the data length—an assumption that is challenged by recent empirical data that demonstrated their meaningful temporal fluctuations (*van Heumen et al., 2021*; *Waschke et al., 2021*). Akin to the motivations behind aperiodic/periodic spectral parameterization and signal decomposition techniques (*Donoghue et al., 2020*; *Wen and Liu, 2016*), undetected temporal variations within the neural spectrogram may conflate fluctuations in aperiodic activity with modulations of periodic signals, hence distorting data interpretation (*Donoghue et al., 2020*). Recent methodological advances have contributed practical tools to decompose and parameterize the neural power spectrum (periodogram) into aperiodic and periodic components (*Donoghue et al., 2020*; *Wen and Liu, 2016*; *He, 2014*). One such practical tool (*specparam*) sequentially fits aperiodic and parametric components to the empirical neural power spectrum (*Donoghue et al., 2020*). The resulting model for the aperiodic component is represented with exponent and offset scalar parameters; periodic elements are modelled with a series of Gaussian-shape functions characterized with three scalar parameters (centre frequency, amplitude, and SD). *Specparam* accounts for static spectral representations and as such does not account for the non-stationary contents of neural time series.

We introduce SPRiNT (Spectral Parameterization Resolved in Time) as a novel approach to identify and model dynamic shifts in aperiodic and periodic brain activity, yielding a time-resolved parameterization of neurophysiological spectrograms. We validate the method with an extensive set of naturalistic simulations of neural data, with benchmark comparisons to parameterized wavelet signal decompositions. Using SPRiNT, we also show that aperiodic fluctuations of the spectrogram can be related to meaningful behavioural and demographic variables from human EEG resting-state data and electrophysiological recordings from free-moving rodents.

## Results

SPRiNT consists of the following methodological steps. First, short-time Fourier transforms (STFTs) are derived from overlapping time windows that slide over data time series. Second, the resulting STFT coefficients are averaged over consecutive time windows to produce a smooth estimate of the power spectral density of the recorded data. Third, the resulting periodogram is parameterized into aperiodic and periodic components with *specparam* (see Methods). As the procedure is repeated over the entire length of the data, SPRiNT produces a time-resolved parameterization of the data's spectrogram (*Figure 1a*). The resulting parameters are then compiled into fully parameterized time-frequency

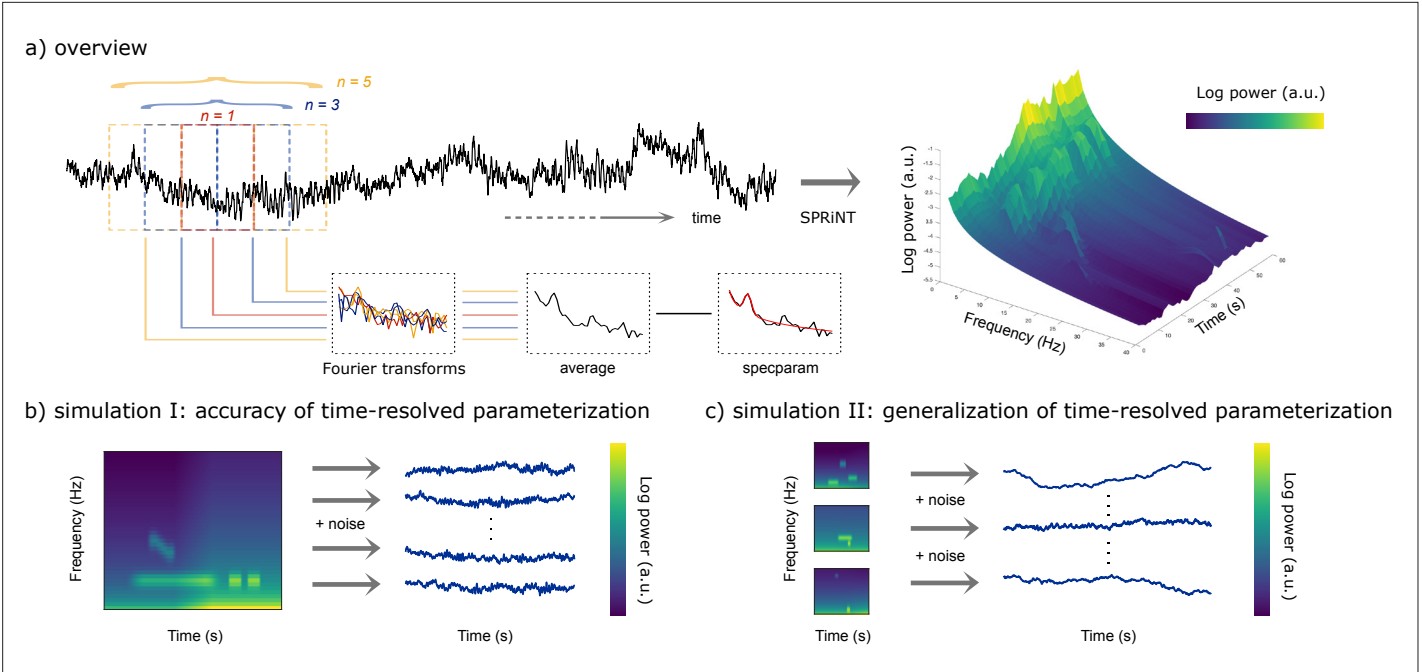

**Figure 1.** Methods synopsis. (**a**) Overview of the Spectral Parameterization Resolved in Time (SPRiNT) approach: At each time bin along a neurophysiological time series (black trace) *n* overlapping time windows are Fourier-transformed to yield an estimate of spectral contents, which is subsequently parameterized using *specparam* (*Donoghue et al., 2020*). The procedure is replicated across time over sliding, overlapping windows to generate a parameterized spectrogram of neural activity. (**b**) Simulation challenge I: We simulated 10,000 time series composed of the same time-varying spectral (aperiodic and periodic) features, with different realizations of additive noise. (**c**) Simulation challenge II: We simulated another 10,000 time series, each composed of different time-varying spectral (aperiodic and periodic) ground-truth features with additive noise. All simulated time series were used to evaluate the respective performances of SPRiNT and the wavelet-*specparam* alternative.

The online version of this article includes the following figure supplement(s) for figure 1:

**Figure supplement 1.** Overview of the outlier peak removal process.

representations for visualization and further derivations. A fourth step consists of an optional post-processing procedure meant to prune outlier transient manifestations of periodic signal components (*Figure 1—figure supplement 1*).

We generated a total of 21,000 naturalistic synthetic neural time series comprising non-stationary aperiodic and periodic signal components, using scripts adapted from the NeuroDSP toolbox (*Cole et al., 2019*) with MATLAB (R2020a; Natick, MA, USA). We first tested SPRiNT's ability to detect and track transient and chirping periodic elements (with time-changing aperiodic components) and benchmarked its performance against parameterized wavelet signal decompositions and parameterized periodograms (*Figure 1b*). A second validation challenge focused on simulations derived from randomly generated sets of realistic aperiodic and periodic parameters; this challenge served to assess SPRiNT's performance across naturalistic heterogeneous signals (*Figure 1c*; see Methods). Further below, we describe the application of SPRiNT to a variety of empirical data from human and rodent electrophysiology.

## Methods benchmarking (synthetic data)

We first simulated 10,000 time series (60 s duration each) with aperiodic components transitioning linearly between t=24 s and t=36 s, from an initial exponent of 1.5 Hz$^{-1}$ and offset of –2.56 (arbitrary units, a.u.) towards a final exponent of 2.0 Hz$^{-1}$ and offset of –1.41 a.u. The periodic components of the simulated signals included transient activity in the alpha band (centre frequency: 8 Hz; amplitude: 1.2 a.u.; SD: 1.2 Hz) occurring between 8 and 40 s, 41–46 s and 47–52 s and a down-chirping oscillation in the beta band centre frequency decreasing from 18 to 15 Hz; amplitude: 0.9 a.u.; SD: 1.4 Hz, between 15 and 25 s (*Figure 1b*). We applied SPRiNT on each simulated time series, post-processed the resulting parameter estimates to remove outlier transient periodic components, and

derived goodness-of-fit statistics of the SPRiNT parameter estimates with respect to their ground-truth values. We compared SPRiNT's performances to parameterized periodograms (*specparam*), as well as the parameterization of temporally smoothed spectrograms obtained from Morlet wavelets time-frequency decompositions of the simulated time series (smoothed using a 4 s Gaussian kernel, SD = 1 s). We refer to the latter approach as wavelet-*specparam* (see Methods). We assessed the respective performances of SPRiNT and wavelet-*specparam* with measures of mean absolute error (MAE) on their respective estimates of aperiodic/periodic spectrogram profiles and of the parameters of their aperiodic/periodic components across time.

Overall, we found that SPRiNT parameterized spectrograms were better fits to ground truth (MAE = 0.04 and SEM = $2.9 \times 10^{-5}$) than those from wavelet-*specparam* (MAE = 0.58 and SEM = $5.1 \times 10^{-6}$; *Figure 2a*). The data showed marked differences in performance between SPRiNT and wavelet-*specparam* in the parameterization of aperiodic components (error on aperiodic spectrogram: wavelet-*specparam* MAE = 0.60 and SEM = $6.7 \times 10^{-6}$; SPRiNT MAE = 0.06 and SEM = $4.0 \times 10^{-5}$). The performances of the two methods in parameterizing periodic components were strong and similar (wavelet-*specparam* MAE = 0.05 and SEM = $4.0 \times 10^{-6}$; SPRiNT MAE = 0.03 and SEM = $2.7 \times 10^{-5}$).

SPRiNT errors on exponents (MAE = 0.11 and SEM = $7.8 \times 10^{-5}$) and offsets (MAE = 0.14 and SEM = $1.1 \times 10^{-4}$) were substantially less than those from wavelet-*specparam* (exponent MAE = 0.19 and SEM = $1.5 \times 10^{-5}$; offset MAE = 0.78 and SEM = $2.6 \times 10^{-5}$; *Figure 2a*). SPRiNT detected periodic alpha activity with higher sensitivity (99% at time bins with maximum peak amplitude) and specificity (96%) than wavelet-*specparam* (95% sensitivity and 47% specificity). SPRiNT estimates of alpha-peak parameters were also closer to ground truth (centre frequency, amplitude, and bandwidth MAE [SEM] = 0.33 [$3.6 \times 10^{-4}$] Hz, 0.20 [$1.7 \times 10^{-4}$] a.u., and 0.42 [$4.8 \times 10^{-4}$] Hz, respectively) than wavelet-*specparam*'s (MAE [SEM]=0.41 Hz [$4.8 \times 10^{-5}$], 0.24 [$2.6 \times 10^{-5}$] a.u., and 0.64 [$6.4 \times 10^{-5}$] Hz, respectively; *Figure 2c*). SPRiNT detected and tracked down-chirping beta periodic activity with higher sensitivity (95% at time bins with maximum peak amplitude) and specificity (98%) than wavelet-*specparam* (62% sensitivity and 90% specificity). SPRiNT's estimates of beta peak parameters were also closer to ground truth (centre frequency, amplitude, and bandwidth MAE = 0.43 [$9.4 \times 10^{-4}$] Hz, 0.17 [$3.6 \times 10^{-4}$] a.u., and 0.48 [$1.1 \times 10^{-3}$] Hz, respectively) than with wavelet-*specparam* (centre frequency, amplitude, and bandwidth MAE = 0.58 [$1.4 \times 10^{-4}$] Hz, 0.16 [$4.2 \times 10^{-5}$] a.u., and 1.05 [$1.2 \times 10^{-4}$] Hz, respectively; *Figure 2c*). We noted that both methods tended to overestimate peak bandwidths (*Figure 2—figure supplement 1*), and the effect was more pronounced with wavelet-*specparam* (*Figure 2c*). While SPRiNT and wavelet-*specparam* performances varied with the chosen parameters (i.e., spectral/temporal resolutions; *Figure 2—figure supplement 2* and *Figure 2—figure supplement 3*; see Supplemental materials), the optimal settings for SPRiNT outperformed the optimal settings for wavelet-*specparam*. We report SPRiNT performances prior to the removal of outlier peaks, as well as wavelet-*specparam* performances without temporal smoothing in Supplemental materials (*Figure 2—figure supplement 4*).

We also parameterized the periodogram of each time series of the first simulation challenge with *specparam* to assess the outcome of a biased assumption of stationary spectral contents across time. The power spectral densities (PSDs) were computed using the Welch approach over 1 s time windows with 50% overlap. The average recovered aperiodic exponent was 1.94 $Hz^{-1}$ (actual = 1.5–2 $Hz^{-1}$) and offset was –1.64 a.u. (actual = –2.56 to –1.41 a.u.). The only peak detected by *specparam* (99% sensitivity) was the alpha peak, with an average centre frequency of 8.09 Hz (actual = 8 Hz), amplitude of 0.79 a.u. (actual max = 1.2 a.u.), and peak frequency SD of 1.21 Hz (actual = 1.2 Hz). No beta peaks were detected across all spectra processed with *specparam*.

## Generalization of SPRiNT across generic aperiodic and periodic fluctuations (synthetic data)

We simulated 10,000 additional time series consisting of aperiodic and periodic components, whose parameters were sampled continuously from realistic ranges (*Figure 1c*). The generators of each trial time series composed: (i) one aperiodic component whose exponent and offset parameters were shifted linearly over time, and (ii) 0–4 periodic components (see Methods for details). SPRiNT, followed by outlier peak post-processing, recovered 69% of the simulated periodic components, with 89% specificity (70% sensitivity and 73% specificity prior to outlier removal as shown in *Figure 3—figure supplement 3*). Aperiodic exponent and offset parameters were recovered with MAEs of 0.12 and

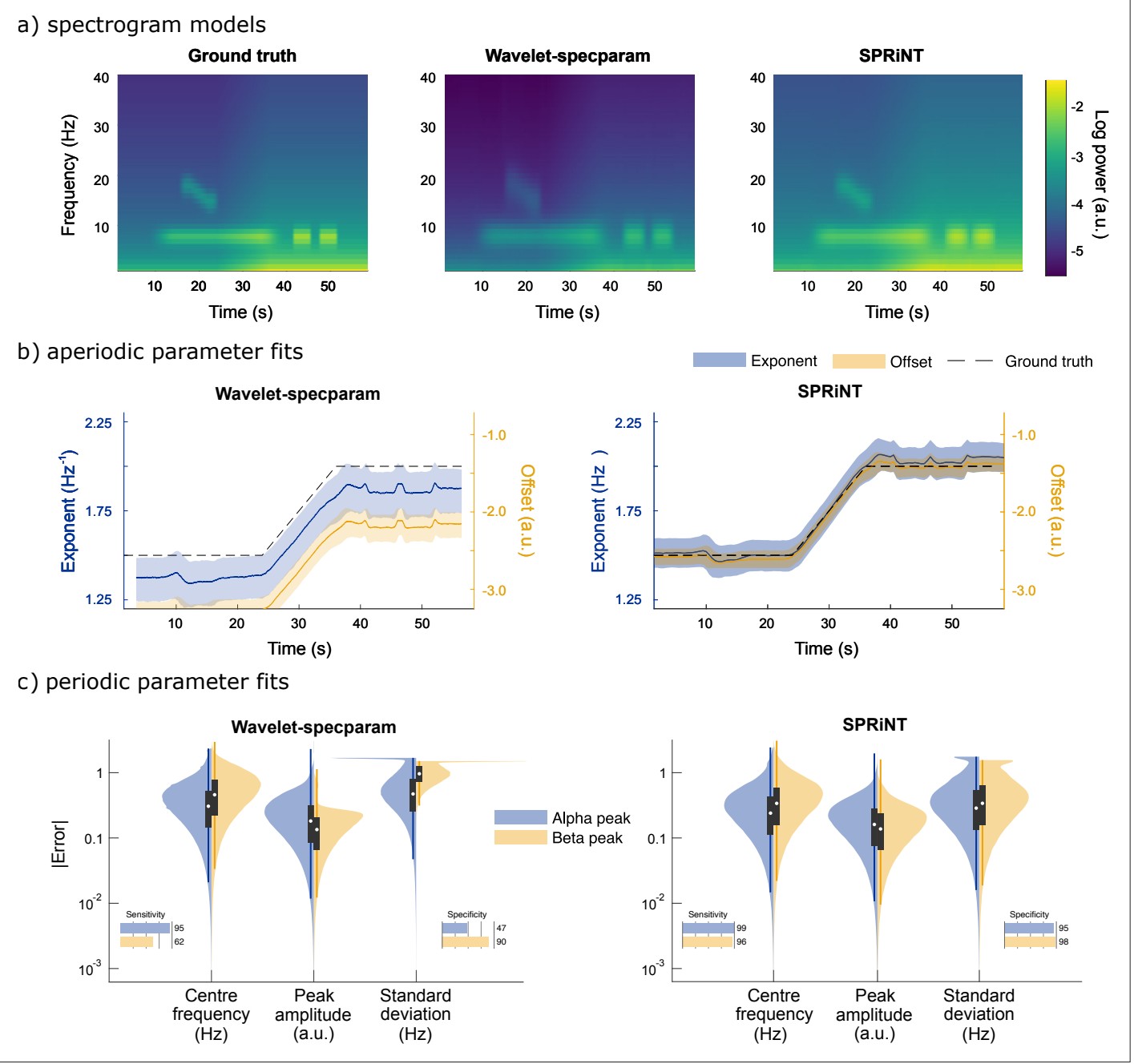

**Figure 2.** Spectral Parameterization Resolved in Time (SPRiNT) vs wavelet-*specparam* performances (simulation challenge I). (**a**) Ground-truth spectrogram (left) and averaged modelled spectrograms produced by the wavelet-*specparam* approach (middle) and SPRiNT (right; n=10,000). (**b**) Aperiodic parameter estimates (lines: median; shaded regions: first and third quartiles, n=10,000) across time from wavelet-*specparam* (left) and SPRiNT (right; black: ground truth; blue: exponent; yellow: offset). (**c**) Absolute error (and detection performance) of alpha and beta-band rhythmic components for wavelet-*specparam* (left) and SPRiNT (right). Violin plots represent the sample distributions (n=10,000; blue: alpha peak; yellow: beta peak; white circle: median, grey box: first and third quartiles; whiskers: range).

The online version of this article includes the following figure supplement(s) for figure 2:

**Figure supplement 1.** Periodic parameter estimates across time.

**Figure supplement 2.** Wavelet-*specparam* performances at varying spectral/temporal resolutions.

**Figure supplement 3.** Spectral Parameterization Resolved in Time (SPRiNT) performances at varying spectral/temporal resolutions.

**Figure supplement 4.** Raw performances of Spectral Parameterization Resolved in Time (SPRiNT) and wavelet-*specparam* (without temporal smoothing and outlier peak removal).

0.15, respectively. The centre frequency, amplitude, and frequency width of periodic components were recovered with MAEs of 0.45, 0.23, and 0.49, respectively (*Figure 3b*). We evaluated whether the detection and accuracy of parameter estimates of periodic components depended on their frequency and amplitude (*Figure 3c*). The synthesized data showed that overall, SPRiNT accurately detects up to two simultaneous periodic components (*Figure 3d*). We also found that periodic components of lower frequencies were more challenging to detect (*Figure 3c,d*; *Figure 3—figure supplement 1b*) because their peak spectral component, when present, tended to be masked by the aperiodic component of the power spectrum. We also observed that lower amplitude peaks were more challenging to detect (*Figure 3c*). However, the detection rate did not depend on peak bandwidth (*Figure 3—figure supplement 1a*). We found that when two or more peaks were present simultaneously, the detection of either or both peaks depended on their spectral proximity (*Figure 3—figure supplement 1c*). Model fit errors (MAE = 0.032) varied significantly with the number of simultaneous periodic components, but this effect was small ($\beta$ = –0.0001, SE = 6.7 × 10$^{-6}$, 95% CI [−0.0001 to 0.0001], p=8.6 × 10$^{-85}$; R$^2$ = 0.0003; *Figure 3e*).

Finally, we simulated 1000 additional time series comprising two periodic components (within the 3–30 Hz and 30–80 Hz ranges, respectively) and a static knee frequency. We used SPRiNT to parameterize the spectrograms of these times series over the 1–100 Hz frequency range (*Figure 3—figure supplement 2*). SPRiNT did not converge to fit aperiodic exponents in the range (–5, 5) Hz$^{-1}$ only on rare occasions (<2% of all time points). We removed these data points from further analysis. The simulated aperiodic exponents and offsets were recovered with MAEs of 0.22 and 0.42, respectively; static knee frequencies were recovered with a MAE of 3.55 × 10$^4$ (inflated by large outliers in absolute error; median absolute error = 11.72). Overall, SPRiNT detected the peaks of the simulated periodic components with 56% sensitivity and 99% specificity. The spectral parameters of periodic components were recovered with equivalent performances in the lower (3–30 Hz) and respectively, higher (30–80 Hz) frequency ranges: MAEs for centre frequency (0.32, resp. 0.32), amplitude (0.27, resp. 0.22), and SD (0.35, resp. 0.29).

## Aperiodic and periodic fluctuations in resting-state EEG dynamics with eyes-closed, and eyes-open behaviour (empirical data)

We applied SPRiNT and *specparam* to resting-state EEG data from the openly available Leipzig Study on Mind-Body-Emotion Interactions (LEMON) dataset (*Babayan et al., 2019*). Participants (n=178) were instructed to open and close their eyes (alternating every 60 s). We used *Brainstorm* (*Tadel et al., 2011*) to preprocess EEG time series from electrode Oz and obtained parameterized spectra with *specparam* and parameterized spectrograms with SPRiNT in both behavioural conditions (eyes open or closed). We also generated time-frequency decompositions of the same preprocessed EEG time series using Morlet wavelets (with default parameters; see Methods and Supplemental materials).

As expected, the group-averaged periodograms showed increased Oz signal power in the alpha range (6–14 Hz) in the eyes-closed behavioural condition with respect to eyes-open (*Figure 4a*). A logistic regression of *specparam* outputs (aperiodic exponent, alpha-peak centre frequency, and alpha-peak amplitude entered as fixed effects) identified alpha-peak power ($\beta$ = –2.73, SE = 0.33, 95% CI [–3.42,–2.11]; Bayes Factor BF = 3.21 × 10$^{-21}$) and aperiodic exponent ($\beta$ = 1.14, SE = 0.42, 95% CI [0.33,–1.99]; BF = 0.20) as predictors of eyes-open or eyes-closed behaviour (*Table 1*). The resulting model suggests that both lower alpha power and larger aperiodic exponents characterize the eyes-open condition.

Using SPRiNT, we found time-varying fluctuations of both aperiodic and alpha-band periodic components as participants opened or closed their eyes (*Figure 4—figure supplement 1*). We observed sharp changes in aperiodic exponent and offset at the transitions between eyes-open and eyes-closed (*Figure 4—figure supplement 1*), which are likely to be artefactual residuals of eye movements. We discarded these segments from further analysis. We ran a logistic regression model with SPRiNT parameter estimates as fixed effects (mean and SD of alpha centre frequency, alpha power, and the aperiodic exponent across time) and found a significant effect of mean alpha power ($\beta$ = –6.31, SE = 0.92, and 95% CI [–8.23,–4.61]), SD of alpha power ($\beta$ = 4.64, SE = 2.03, and 95% CI [0.76, 8.73]), and mean aperiodic exponent ($\beta$ = 2.55, SE = 0.53, and 95% CI [1.55, 3.63]) as predictors of the behavioural condition (*Table 2*). According to this model, lower alpha power, larger aperiodic exponents, and stronger fluctuations of alpha-band activity over time are signatures of the eyes-open

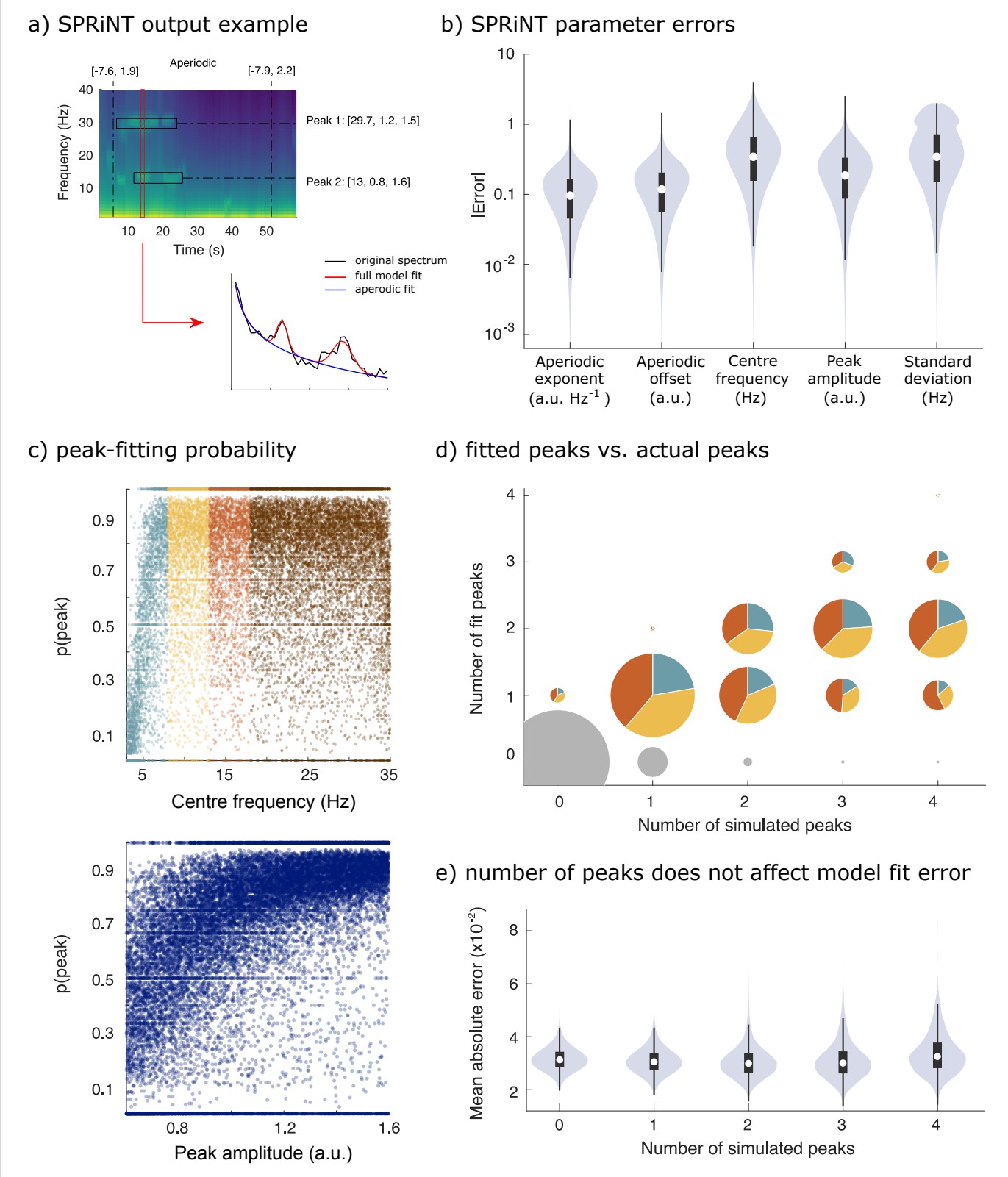

**Figure 3.** Spectral Parameterization Resolved in Time (SPRiNT) performances (simulation challenge II). (**a**) SPRiNT parameterized spectrogram for a representative simulated time series with time-varying aperiodic (offset and exponent) and transient periodic (centre frequency, amplitude, and SD) components. The red arrow indicates a cross-sectional view of the spectrogram at 14 s. (**b**) Absolute error in SPRiNT parameter estimates across all simulations (n=10,000). (**c**) Detection probability of spectral peaks (i.e., rhythmic components) depending on simulated centre frequency and amplitude

*Figure 3 continued on next page*

*Figure 3 continued*

(light blue: 3–8 Hz theta; yellow: 8–13 Hz alpha; orange: 13–18 Hz beta; brown:18–35 Hz). (**d**) Number of fitted vs simulated periodic components (spectral peaks) across all simulations and time points. The underestimation of the number of estimated spectral peaks is related to centre frequency: 3–8 Hz simulated peaks (light blue) account for proportionally fewer of recovered peaks between 3 and 18 Hz (light blue, yellow, and orange) than from the other two frequency ranges. Samples sizes by number of simulated peaks: 0 peaks = 798,753, 1 peak = 256,599, 2 peaks = 78,698, 3 peaks = 14,790, 4 peaks = 1160. (**e**) Model fit error is not affected by number of simulated peaks. Violin plots represent the full sample distributions (white circle: median, grey box: first and third quartiles; whiskers: range).

The online version of this article includes the following source data and figure supplement(s) for figure 3:

**Source data 1.** Figure data for simulation challenge II.

**Figure supplement 1.** Performances of Spectral Parameterization Resolved in Time (SPRiNT) across a range of peak SD, frequency band, and spectral separation between peaks.

**Figure supplement 2.** Performances of Spectral Parameterization Resolved in Time (SPRiNT) on broad-range spectrograms comprising spectral knees.

**Figure supplement 3.** Performances of Spectral Parameterization Resolved in Time (SPRiNT) (without outlier peak removal).

resting condition. A Bayes factor analysis confirmed the evidence of effects of mean alpha power (BF = $4.39 \times 10^{-13}$) and mean aperiodic exponent (BF = $1.62 \times 10^{-4}$), and indicated mild evidence against the temporal variability of alpha power (BF = 3.81; *Table 2*). Although model fit error was slightly higher in the eyes-closed condition, it did not affect condition relationships when included in a logistic regression (see Supplemental materials; *Table 3*). In summary, both *specparam* and SPRiNT analyses confirmed alpha power and aperiodic exponent as neurophysiological markers of eyes-closed vs eyes-open behaviour. Wavelet analyses confirmed that mean alpha-band activity predicted behavioural condition ($\beta = -2.05$, SE = 0.31, and 95% CI [–2.67,–1.47]; BF = $1.08 \times 10^{-11}$; *Table 4*). We emphasize that SPRiNT's spectrogram parameterization was uniquely able to reveal time-varying changes in alpha power related to eyes-closed vs eyes-open behaviour, although the Bayes factor for this effect suggests it to be marginal.

## Prediction of biological age group from aperiodic and periodic components of the resting-state EEG spectrogram (empirical data)

Using the same dataset, we tested the hypothesis that SPRiNT parameter estimates are associated with participants' age group (i.e., younger [n=121] vs older [n=57] adults). Extant literature reports slower alpha rhythms and smaller aperiodic exponents in healthy ageing (*Donoghue et al., 2020*). We performed a logistic regression based on SPRiNT parameter estimates of the mean and SD of alpha centre frequency, alpha power, and aperiodic exponent as fixed effects in the eyes-open condition. We found significant effects of mean aperiodic exponent ($\beta = -3.31$, SE = 0.75, and 95% CI [−4.88,– 1.91]) and SD of alpha centre frequency ($\beta = 1.30$, SE = 0.53, and 95% CI [0.28, 2.39]; *Table 5*). We therefore found using SPRiNT that the EEG spectrogram of older participants decreased less rapidly with frequency (characterized by a smaller exponent) and revealed stronger time-varying fluctuations of alpha-peak centre frequency. A Bayes factor analysis showed strong evidence for the effect of the aperiodic exponent (BF = $5.14 \times 10^{-5}$) and for the variability of the alpha-peak centre frequency (BF = 0.20; *Table 5*).

We replicated the same SPRiNT parameter analysis with the data in the eyes-closed condition. We found that mean aperiodic exponent ($\beta = -4.34$, SE = 0.84, and 95% CI [−6.10,–2.79]) and mean alpha centre frequency ($\beta = -0.74$, SE = 0.27, and 95% CI [−1.28,–0.24]) were predictors of participants' age group, with older participants again showing a flatter spectrum and a slower alpha peak (lower centre frequency; *Table 6*). A Bayes factor analysis provided strong evidence for the effect of mean aperiodic exponent (BF = $1.10 \times 10^{-7}$) and for the effect of mean alpha centre frequency (BF = 0.07; *Table 6*).

We performed an additional logistic regression to predict age group using the mean and variability (SD) of individual alpha-peak frequency (between 6 and 14 Hz) from the STFT as fixed effects. We found that only variability in eyes-open individual alpha-peak frequency predicted age group ($\beta = 0.63$, SE = 0.30, and 95% CI [0.04, 1.24]), though a Bayes factor analysis showed anecdotal evidence for this effect (BF = 0.59; *Table 7*, see also *Table 8*). Measures of individual alpha-peak frequency can be distorted by aperiodic activity (*Donoghue et al., 2020*) and by the absence of a clear peak in the spectrum. In that regard, SPRiNT can help clarify the underlying dynamical structure of the observed

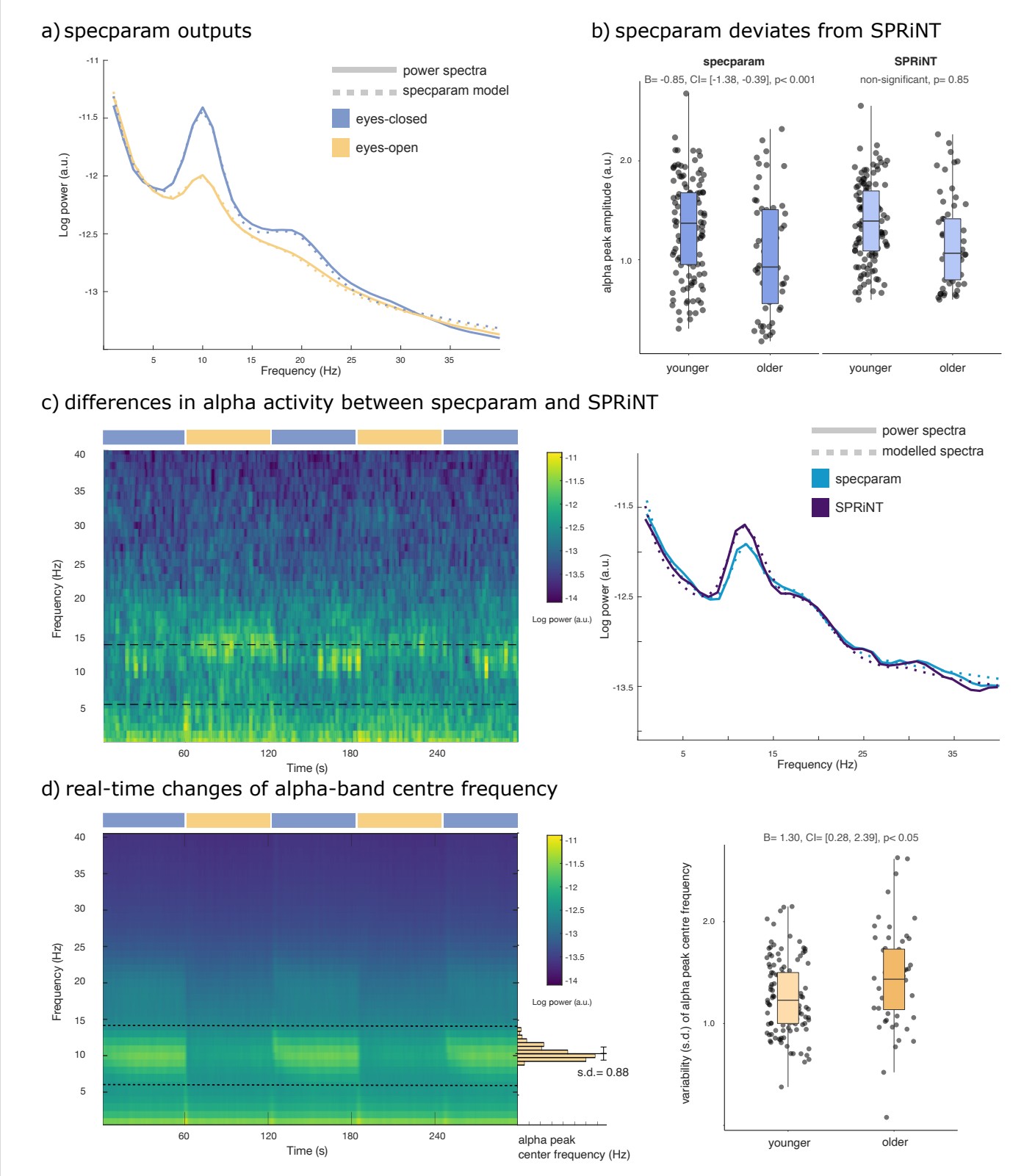

**Figure 4.** Spectral Parameterization Resolved in Time (SPRiNT) parameterization of resting-state EEG. (**a**) Mean periodogram and *specparam* models for eyes-closed (blue) and eyes-open (yellow) resting-state EEG activity (from electrode Oz; n=178). (**b**) Logistic regressions showed that *specparam*-derived eyes-closed alpha-peak amplitude was predictive of age group, but mean eyes-closed alpha-peak amplitude derived from SPRiNT was not. (**c**) Example of intrinsic dynamics in alpha activity during the eyes-closed period leading to divergent SPRiNT and *specparam* models (participant sub-

*Figure 4 continued on next page*

*Figure 4 continued*

016). In a subset of participants (<10%), we observed strong intermittence of the presence of an alpha peak. Since an alpha peak was not consistently present in the eyes-closed condition, and *specparam*-derived alpha-peak amplitude (0.77 a.u.; light blue) is lower than SPRiNT-derived mean alpha-peak amplitude (1.06 a.u.; dark blue), as the latter only includes time samples featuring a detected alpha peak. (**d**) Logistic regression showed that temporal variability in eyes-open alpha centre frequency predicts age group. Left: mean SPRiNT spectrogram (n=178) and sample distribution of eyes-open alpha centre frequency (participant sub-067). Right: variability (SD) in eyes-open alpha centre frequency separated by age group. Note: no alpha peaks were detected in the eyes-open period for one participant (boxplot line: median; boxplot limits: first and third quartiles; whiskers: range). Sample sizes: younger adults (age: 20–40 years): 121; older adults (age: 55–80 years): 56.

The online version of this article includes the following source data and figure supplement(s) for figure 4:

**Source data 1.** Spectral parameters and age group by participant.

**Figure supplement 1.** Spectral Parameterization Resolved in Time (SPRiNT) model parameters in resting-state EEG.

effects by systematically decomposing spectrograms into explicitly detected time-varying aperiodic and periodic components.

We also tested whether the observed differences in mean spectral parameters could be replicated from the parameterization of the periodograms using *specparam*. We performed a logistic regression based on *specparam* parameter estimates of alpha centre frequency, alpha power, and aperiodic exponent as fixed effects from the average periodogram, in both behavioural conditions. We confirmed significant effects in all the same predictors as detected by SPRiNT: eyes-open aperiodic exponent ($\beta$ = –3.30, SE = 0.85, and 95% CI [–5.08,–1.74]; *Table 9*), eyes-closed aperiodic exponent ($\beta$ = –2.67, SE = 0.61, and 95% CI [–3.94,–1.54]), and eyes-closed alpha centre frequency ($\beta$ = –0.85, SE = 0.25, and 95% CI [–1.38,–0.39]; *Table 10*). However, we found significant effects for eyes-open alpha centre frequency ($\beta$ = –0.35, SE = 0.16, and 95% CI [–0.68,–0.05]; *Table 9*) and eyes-closed alpha power ($\beta$ = –0.96, SE = 0.37, and 95% CI [–1.72,–0.24]; *Table 10*), which were not observed using SPRiNT (*Figure 4b*). We also observed intrinsic dynamics in the alpha band of a subset of participants (<10%) contributing to diverging measures of alpha-peak amplitude between *specparam* and SPRiNT (*Figure 4c*).

Finally, we performed a logistic regression using mean alpha power from the wavelet spectrogram as a fixed effect and found that mean alpha power discriminated between age groups only in the eyes-closed condition ($\beta$ = –1.13, SE = 0.38, and 95% CI [–1.90 to 0.41]; *Table 11*; see also *Table 12*). Because wavelet spectrograms are not readily decomposed into aperiodic and periodic components, these findings may be biased by age-related effects on aperiodic exponent, alpha-peak centre frequency (*Scally et al., 2018*), and the absence of an actual periodic component in the alpha range.

## Transient changes in aperiodic brain activity are associated with locomotor behaviour (empirical data)

We used intracranial data from two Long-Evans rats recorded in layer 3 of entorhinal cortex while they moved freely along a linear track (*Mizuseki et al., 2009*; https://crcns.org). Rats travelled alternatively to either end of the track to receive a water reward, resulting in behaviours of recurring bouts of running and resting. Power spectral density estimates revealed substantial broadband power increases below 20 Hz during rest relative to movement (except for spectral peaks around 8 Hz and

**Table 1.** Logistic regression model of *specparam* parameters for predicting condition (eyes-closed vs eyes-open).

| *Predictors* | Condition | | | |
| :--- | :--- | :--- | :--- | :--- |
| | **Log-Odds** | **CI** | **p** | **BF** |
| (Intercept) | 0.86 | –1.85–3.64 | 0.537 | |
| Alpha centre frequency (*specparam*) | 0.00 | –0.23–0.23 | 0.990 | 7.97 |
| Alpha amplitude (*specparam*) | –2.73 | –3.42 to –2.11 | **<0.001** | 3.21 e-21 |
| Aperiodic exponent (*specparam*) | 1.14 | 0.33–1.99 | **0.007** | 0.20 |
| Observations | 323 | | | |
| R² Tjur | 0.284 | | | |

**Table 2.** Logistic regression model of Spectral Parameterization Resolved in Time (SPRiNT) parameters for predicting condition (eyes-closed vs eyes-open).

| Predictors | Condition | | | |
| --- | --- | --- | --- | --- |
| | Log-Odds | CI | p | BF |
| (Intercept) | 0.10 | −3.75–4.02 | 0.959 | |
| Mean alpha centre frequency | 0.24 | −0.04–0.52 | 0.101 | 1.58 |
| Std alpha centre frequency | −0.06 | −0.97–0.86 | 0.898 | 4.39 |
| Mean alpha power | −6.31 | −8.23 to −4.61 | <0.001 | 4.51e-13 |
| Std alpha power | 4.64 | 0.76–8.73 | 0.022 | 3.81 |
| Mean aperiodic slope | 2.55 | 1.55–3.63 | <0.001 | 1.62e-4 |
| Std aperiodic slope | −2.74 | −8.54–3.38 | 0.362 | 4.32 |
| Observations | 355 | | | |
| $R^2$ Tjur | 0.432 | | | |

harmonics; *Figure 5*; *Samiee and Baillet, 2017*). We therefore tested for the possible expression of two alternating modes of aperiodic neural activity associated with each behaviour. SPRiNT parameterization found in the two subjects that resting bouts were associated with larger aperiodic exponents and more positive offsets than during movement bouts (*Figure 5—figure supplement 2*). We ran SPRiNT parameterizations over 8 s epochs proximal to transitions between movement and rest; we observed dynamic shifts between aperiodic modes associated with behavioural changes (*Figure 5*). We tested whether changes in aperiodic exponent proximal to transitions of movement and rest were related to movement speed and found a negative linear association in both subjects for both transition types (EC012 transitions to rest: $\beta = -9.6 \times 10^{-3}$, SE = $4.7 \times 10^{-4}$, 95% CI [$-1.1 \times 10^{-2}$ $-8.6 \times 10^{-3}$], p<0.001, $R^2$ = 0.29; EC012 transitions to movement: $\beta = -7.3 \times 10^{-3}$, SE = $4.3 \times 10^{-4}$, 95% CI [$-8.1 \times 10^{-3}$ $-6.4 \times 10^{-3}$], p<0.001, $R^2$ = 0.18; EC013 transitions to rest: $\beta = -1.1 \times 10^{-2}$, SE = $2.3 \times 10^{-4}$, 95% CI [$-1.2 \times 10^{-2}$ $-1.1 \times 10^{-2}$], p<0.001, $R^2$ = 0.32; EC013 transitions to movement: $\beta = -1.2 \times 10^{-2}$, SE = $3.2 \times 10^{-4}$, 95% CI [$-1.3 \times 10^{-2}$ $-1.2 \times 10^{-2}$], p<0.001, $R^2$ = 0.26; *Figure 5—figure supplement 3*). We emphasize that the periodic features of the recordings were non-sinusoidal and therefore were not explored further with the methods discussed herein (*Donoghue et al., 2021*; *Figure 5—figure supplement 1*).

**Table 3.** Logistic regression model of Spectral Parameterization Resolved in Time (SPRiNT) parameters for predicting condition (eyes-closed vs eyes-open), with model fit error (mean absolute error [MAE]) as a predictor.

| Predictors | Condition | | |
| --- | --- | --- | --- |
| | Log-Odds | CI | p |
| (Intercept) | −1.37 | −8.83–4.07 | 0.620 |
| Mean alpha centre frequency | 0.23 | −0.05–0.51 | 0.115 |
| Std alpha centre frequency | −0.15 | −1.08–0.79 | 0.751 |
| Mean alpha power | −6.62 | −8.73 to −4.73 | <0.001 |
| Std alpha power | 5.15 | 1.05–9.46 | 0.016 |
| Mean aperiodic slope | 2.63 | 1.60–3.73 | <0.001 |
| Std aperiodic slope | −3.79 | −10.21–2.89 | 0.253 |
| Model fit MAE | 59.96 | −95.00–215.29 | 0.447 |
| Observations | 355 | | |
| $R^2$ Tjur | 0.433 | | |

**Table 4.** Logistic regression model parameters for predicting condition (eyes-closed vs eyes-open) from Morlet wavelet spectrograms.

| Predictors | Condition | | | |
| | Log-Odds | CI | p | BF |
|---|---|---|---|---|
| (Intercept) | −25.98 | −33.87 to −18.65 | <0.001 | |
| Alpha power (Morlet wavelets) | −2.05 | −2.67 to −1.47 | <0.001 | 1.08e-11 |
| Observations | 356 | | | |
| R² Tjur | 0.148 | | | |

## Discussion

We introduce SPRiNT as a new method to parameterize dynamic fluctuations in the spectral contents of neurophysiological time series. SPRiNT extends recent practical tools that determine aperiodic and periodic parameters from static power spectra of neural signals to their spectrograms. Aperiodic spectral components may confound the detection and interpretation of narrow-band power changes as periodic, oscillatory signal elements. Given the scientific prominence of measures of neural oscillations in (causal) relation to behaviour (e.g., *Albouy et al., 2017*) and clinical syndromes (e.g., *Ostlund et al., 2021*), it is essential that their characterization in time and frequency be contrasted with that of the underlying aperiodic background activity at the natural time scale of behaviour and perception.

### SPRiNT expands the neural spectrogram toolkit

Recent empirical studies show that the spectral distribution of neural signal power with frequency can be decomposed into low-dimensional aperiodic and periodic components (*Donoghue et al., 2020*) and that these latter are physiologically (*Cole et al., 2019*), clinically (*Molina et al., 2020*; *van Heumen et al., 2021*), and behaviourally (*Ouyang et al., 2020*; *Waschke et al., 2021*) meaningful.

SPRiNT extends the approach to the low-dimensional time-resolved parameterization of neurophysiological spectrograms. The method combines the simplicity of the *specparam* spectral decomposition approach with the computational efficiency of STFTs across sliding windows. The present results demonstrate its technical concept and indicate that SPRiNT unveils meaningful additional information from the data beyond established tools such as wavelet time-frequency decompositions.

Using realistic simulations of neural time series, we demonstrate the strengths and current limitations of SPRiNT. We show that SPRiNT decompositions provide a comprehensive account of the neural spectrogram (*Figure 2a*), tracking the dynamics of periodic and aperiodic signal components across time (*Figure 2b* and *Figure 2—figure supplement 1*). We note that the algorithm performs optimally when the data features narrow-band oscillatory components that can be characterized as

**Table 5.** Eyes-open logistic regression model parameters for predicting age group, Spectral Parameterization Resolved in Time (SPRiNT).

| Predictors | Age | | | |
| | Log-Odds | CI | p | BF |
|---|---|---|---|---|
| (Intercept) | 1.92 | −2.82–6.80 | 0.428 | |
| Eyes-open mean alpha centre frequency | −0.05 | −0.39–0.29 | 0.789 | 3.43 |
| Eyes-open std alpha centre frequency | 1.30 | 0.28–2.39 | 0.015 | 0.20 |
| Eyes-open mean alpha power | 0.41 | −2.69–3.27 | 0.784 | 2.97 |
| Eyes-open std alpha power | −3.81 | −9.47–1.54 | 0.172 | 1.14 |
| Eyes-open mean aperiodic slope | −3.31 | −4.88 to −1.91 | <0.001 | 5.14e-05 |
| Eyes-open std aperiodic slope | 3.44 | −4.83–11.06 | 0.388 | 2.66 |
| Observations | 177 | | | |
| R² Tjur | 0.216 | | | |

**Table 6.** Eyes-closed logistic regression model parameters for predicting age group, Spectral Parameterization Resolved in Time (SPRiNT).

| | Age | | | |
| Predictors | Log-Odds | CI | p | BF |
| --- | --- | --- | --- | --- |
| (Intercept) | 11.23 | 4.63–18.50 | 0.001 | |
| Eyes-closed mean centre frequency | –0.74 | –1.28 to –0.24 | 0.006 | 0.07 |
| Eyes-closed std centre frequency | 1.01 | –0.48–2.56 | 0.188 | 1.65 |
| Eyes-closed mean alpha power | –0.15 | –1.76–1.43 | 0.852 | 3.90 |
| Eyes-closed std alpha power | –0.51 | –5.32–4.22 | 0.831 | 3.61 |
| Eyes-closed mean aperiodic slope | –4.34 | –6.10 to –2.79 | <0.001 | 1.10e-07 |
| Eyes-closed std aperiodic slope | 0.54 | –9.66–9.45 | 0.910 | 3.93 |
| Observations | 178 | | | |
| $R^2$ Tjur | 0.272 | | | |

spectral peaks (*Figure 3c*). The algorithm performs best when the data contains two or fewer periodic components concurrently (*Figure 3d*). We found that these current limitations are inherent to *specparam*, which is challenged by the dissociation of spectral peaks from background aperiodic activity at the lower edge of the power spectrum (*Donoghue et al., 2020*).

Our synthetic data also identified certain limitations of the SPRiNT approach. The algorithm tends to overestimate the bandwidth of spectral peaks, which we discuss as related to the frequency resolution of the spectrogram (mostly 1 Hz in the present study). The frequency resolution of the spectrogram at 1 Hz, e.g., may be too low to quantify narrower band-limited components. The intrinsic noise level present in STFTs (i.e., spectral power not explained by periodic or aperiodic components) may also challenge bandwidth estimation. Increasing STFT window length augments spectral resolution and reduces intrinsic noise, although to the detriment of temporal specificity. We also found that SPRiNT may underestimate the number of periodic components, though this can be interpreted as the joint probability of SPRiNT detecting multiple independent oscillatory peaks (where the probability of detecting a given peak is between 65 and 75%; approximating a binomial distribution). We found that a peak was more likely to be detected if its amplitude is stronger and the centre frequency is above 8 Hz (*Figure 3c* and *Figure 3—figure supplement 1b*), and if separated from other peaks by at least 8 Hz (*Figure 3—figure supplement 1d*). Finally, SPRiNT's performances were slightly degraded when spectrograms composed an aperiodic knee (*Figure 3—figure supplement 2*). This is due to the specific challenge of estimating knee parameters. Nevertheless, the spectral knee frequency is related to intrinsic neuronal timescales and cortical microarchitecture (*Gao et al., 2020*), which are expected to be stable properties within each individual and across a given recording. Thus, we recommend estimating (and reporting) aperiodic knee frequencies from the power spectrum of the data with *specparam* and specifying the estimated value as a SPRiNT parameter.

**Table 7.** Eyes-open logistic regression model parameters for predicting age group, short-time Fourier transform (STFT).

| | Age | | | |
| Predictors | Log-Odds | CI | p | BF |
| --- | --- | --- | --- | --- |
| (Intercept) | –0.44 | –3.04–2.11 | 0.734 | |
| Eyes-open mean individual alpha-peak frequency (STFT) | –0.17 | –0.45–0.11 | 0.233 | 2.33 |
| Eyes-open std individual alpha-peak frequency (STFT) | 0.63 | 0.04–1.24 | 0.040 | 0.59 |
| Observations | 178 | | | |
| $R^2$ Tjur | 0.026 | | | |

**Table 8.** Eyes-closed logistic regression model parameters for predicting age group, short-time Fourier transform (STFT).

| Predictors | Age | | | |
| --- | --- | --- | --- | --- |
| | Log-Odds | CI | p | BF |
| (Intercept) | 1.83 | −1.98–5.75 | 0.350 | |
| Eyes-closed mean individual alpha-peak frequency (STFT) | −0.31 | −0.70–0.07 | 0.113 | 1.28 |
| Eyes-closed std individual alpha-peak frequency (STFT) | 0.30 | −0.22–0.81 | 0.256 | 2.32 |
| Observations | 178 | | | |
| $R^2$ Tjur | 0.024 | | | |

SPRiNT's optional outlier peak removal procedure increases the specificity of detected spectral peaks by emphasizing the detection of periodic components that develop over time. This feature is controlled by threshold parameters that can be adjusted along the time and frequency dimensions. So far, we found that applying a semi-conservative threshold for outlier removal (i.e., if less than three more peaks are detected within 2.5 Hz and 3 s around a given peak of the spectrogram) reduced the false detection rate by 50%, without affecting the true detection rate substantially (a<5% reduction; *Figure 3* and *Figure 3—figure supplement 3*). Setting these threshold parameters too conservatively would reduce the sensitivity of peak detection.

Practical mitigation techniques have been proposed to account for the presence of background aperiodic activity when estimating narrow-band signal power changes. For instance, baseline normalization is a common approach used to isolate event-related signals and prepare spectrograms for comparisons across individuals (*Cohen, 2014*). However, the resulting relative measures of event-related power increases or decreases do not explicitly account for the fact that behaviour or stimulus presentations may also induce rapid changes in aperiodic activity. Therefore, baseline normalization followed by narrow-band analysis of power changes is not immune to interpretation ambiguities when aperiodic background activity also changes dynamically. Further, the definition of a reference baseline can be inadequate for some study designs, as exemplified herein with the LEMON dataset.

## SPRiNT decomposition of EEG data tracks and predicts behaviour and demographics

We found in the LEMON dataset that measuring narrow band power changes without accounting for concurrent variations of the aperiodic signal background challenges the interpretation of effects manifested in the spectrogram (*Scally et al., 2018*). Spectral parameterization with SPRiNT or *specparam* enables this distinction, showing that both periodic and aperiodic changes in neural activity are associated with age and behaviour. We found strong evidence for decreases in alpha-peak power and increases in aperiodic exponent during eyes-open resting-state behaviour (compared to eyes-closed; *Figure 4a*). However, it remains unclear whether these effects are independent or related. A recent analysis of the same dataset showed that the amplitude of alpha oscillations around a non-zero mean voltage influences baseline cortical excitability (*Studenova et al., 2021*)—an effect observable in part

**Table 9.** Eyes-open logistic regression model parameters for predicting age group, *specparam*.

| Predictors | Age | | | |
| --- | --- | --- | --- | --- |
| | Log-Odds | CI | p | BF |
| (Intercept) | 7.61 | 3.63–12.09 | <0.001 | |
| Eyes-open aperiodic exponent (*specparam*) | −3.30 | −5.08 to −1.74 | <0.001 | 4.61 e-4 |
| Eyes-open alpha centre frequency (*specparam*) | −0.35 | −0.68 to −0.05 | 0.028 | 0.26 |
| Eyes-open alpha amplitude (*specparam*) | −1.34 | −2.86–0.02 | 0.066 | 0.72 |
| Observations | 147 | | | |
| $R^2$ Tjur | 0.207 | | | |

**Table 10.** Eyes-closed logistic regression model parameters for predicting age group, *specparam*.

| Predictors | Age | | | |
| | Log-Odds | CI | p | BF |
| --- | --- | --- | --- | --- |
| (Intercept) | 12.40 | 7.11–18.50 | <0.001 | |
| Eyes-closed aperiodic exponent (*specparam*) | –2.67 | –3.94 to –1.54 | <0.001 | 3.22e-5 |
| Eyes-closed alpha centre frequency (*specparam*) | –0.85 | –1.38 to –0.39 | 0.001 | 3.61e-3 |
| Eyes-closed alpha amplitude (*specparam*) | –0.96 | –1.72 to –0.24 | 0.010 | 0.11 |
| Observations | 176 | | | |
| $R^2$ Tjur | 0.246 | | | |

through variations of the aperiodic exponent (*Gao et al., 2017*). Using both SPRiNT and *specparam*, we also observed both slower alpha-peak centre frequencies and smaller aperiodic exponents in the older age group, in agreement with previous literature on healthy ageing (*Cellier et al., 2021*; *Donoghue et al., 2020*; *Hill et al., 2022*; *Ostlund et al., 2022*; *Schaworonkow and Voytek, 2021*).

Using *specparam*, we found lower alpha-band peak amplitudes in older individuals in the eyes-closed condition. We could not replicate this effect from spectrograms parameterized with SPRiNT (*Figure 4b*). This apparent divergence may be due to the challenge of detecting low-amplitude peaks in the spectrogram of older individuals. Periodograms are derived from averaging across time windows, which augments signal-to-noise ratios (SNRs), and therefore the sensitivity of *specparam* to periodic components of lower amplitude. In a subset of participants (<10%), we also observed considerable differences between the alpha-peak amplitudes extracted from *specparam* and SPRiNT, which we explained by unstable expressions of alpha activity over time in these participants (*Figure 4c*). The average alpha-peak amplitude estimated with SPRiNT is based only on time segments when an alpha-band periodic component is detected. With *specparam*, this estimate is derived across all time windows, regardless of the presence/absence of a bona fide alpha component at certain time instances. The consequence is that the estimate of the average alpha-peak amplitude is larger with SPRiNT than with *specparam* in these participants. Therefore, differences in alpha power between SPRiNT and *specparam* may be explained, at least in some participants, by differential temporal fluctuations of alpha band activity (*Donoghue et al., 2021*). This effect is reminiscent of recent observations that beta-band power suppression during motor execution is due to sparser bursting activity, not a sustained decrease of beta-band activity (*Sherman et al., 2016*).

We also emphasize how the variability of spectral parameters may relate to demographic features, as shown with SPRiNT's prediction of participants' age from the temporal variability of eyes-open alpha-peak centre frequency (*Figure 4d*). This could account for the interpretation derived from the periodogram, where eyes-open alpha-peak centre frequency is predictive of age instead. Previous studies explored similar effects of within-subject variability of alpha-peak centre frequency (*Haegens et al., 2014*) and their clinical relevance (*Larsson and Kostov, 2005*). These findings augment the recent evidence that neural spectral features are robust signatures proper to an individual (*da Silva Castanheira et al., 2021*) and open the possibility that their temporal variability is neurophysiologically significant.

**Table 11.** Eyes-closed logistic regression model parameters for predicting age group, Morlet wavelets.

| Predictors | Age | | | |
| | Log-Odds | CI | p | BF |
| --- | --- | --- | --- | --- |
| (Intercept) | –14.93 | –24.68 to –5.89 | 0.002 | |
| Eyes-closed alpha power (Morlet wavelets) | –1.13 | –1.90 to –0.41 | 0.003 | 0.07 |
| Observations | 178 | | | |
| $R^2$ Tjur | 0.053 | | | |

**Table 12.** Eyes-open logistic regression model parameters for predicting age group, Morlet wavelets.

| Predictors | Age | | | |
| --- | --- | --- | --- | --- |
| | Log-Odds | CI | p | BF |
| (Intercept) | −9.04 | −21.73–3.13 | 0.152 | |
| Eyes-open alpha power (Morlet wavelets) | −0.64 | −1.63–0.30 | 0.189 | 2.74 |
| Observations | 178 | | | |
| $R^2$ Tjur | 0.010 | | | |

We also report time-resolved fluctuations in aperiodic activity related to behaviour in freely moving rats (*Figure 5*). SPRiNT aperiodic parameters highlight larger spectral exponents in rats during rest than during movement. Time-resolved aperiodic parameters can also be tracked with SPRiNT as subjects transition from periods of movement to rest and vice versa. The smaller aperiodic exponents observed during movement may be indicative of periods of general cortical disinhibition (*Gao et al., 2017*). Previous work on the same data has shown how locomotor behaviour is associated with changes in amplitude and centre frequency of entorhinal theta rhythms (*Mizuseki et al., 2009*; *Samiee and Baillet, 2017*). We also note that strong theta activity may challenge the estimation of aperiodic parameters (*Gao et al., 2017*). Changes in aperiodic exponent were partially explained by movement speed (*Figure 5—figure supplement 3*), which could reflect increased processing demands from additional spatial information entering entorhinal cortex (*Keene et al., 2016*) or increased activity in cells encoding speed directly (*Iwase et al., 2020*). Combined, the reported findings support the notion that aperiodic background neural activity changes in relation to a variety of contexts and subject types (*Donoghue et al., 2020*; *Gao et al., 2017*; *Molina et al., 2020*; *Ostlund et al., 2021*; *Pathania et al., 2021*; *Waschke et al., 2021*; *van Heumen et al., 2021*). *Gao et al., 2017* established a link between aperiodic exponent and the local balance of neural excitation vs inhibition. How this balance adjusts dynamically, potentially over a multiplicity of time scales, and relates directly or indirectly to individual behaviour, demographics, and neurophysiological factors remains to be studied.

## Practical recommendations for using SPRiNT

SPRiNT returns goodness-of-fit metrics for all spectrogram parameters. However, these metrics cannot account entirely for possible misrepresentations or omissions of certain components of the spectrogram. Visual inspections of original spectrograms and SPRiNT parameterizations are recommended, e.g., to avoid fitting a 'fixed' aperiodic model to data with a clear spectral knee or to ensure that the minimum peak height parameter is adjusted to the peak of lowest amplitude in the data. Most of the results presented here were obtained with similar SPRiNT parameter settings. Below are practical recommendations for SPRiNT parameter settings, in mirror and complement of those provided by *Ostlund et al., 2022* and *Gerster et al., 2022* for *specparam*:

- Window length determines the frequency and temporal resolution of the spectrogram. This parameter needs to be adjusted to the expected timescale of the effects under study so that multiple overlapping SPRiNT time windows cover the expected duration of the effect of interest; see for instance, the 2 s time windows with 75% overlap designed to detect the effect at the timescale characterized in *Figure 5*.
- Window overlap ratio is a companion parameter of window length that also determines the temporal resolution of the spectrogram. While a greater overlap ratio increases the rate of temporal sampling, it also increases the redundancy of the data information collected within each time window and therefore smooths the spectrogram estimates over the time dimension. A general recommendation is that longer time windows (>2 s) enable larger overlap ratios (>75%). We recommend a default setting of 50% as a baseline for data exploration.
- Number of windows averaged in each time bin enables to control the SNR of the spectrogram estimates (higher SNR with more windows averaged), with the companion effect of increasing the temporal smoothing (i.e., decreased temporal resolution) of the spectrogram. We recommend a baseline setting of five windows.

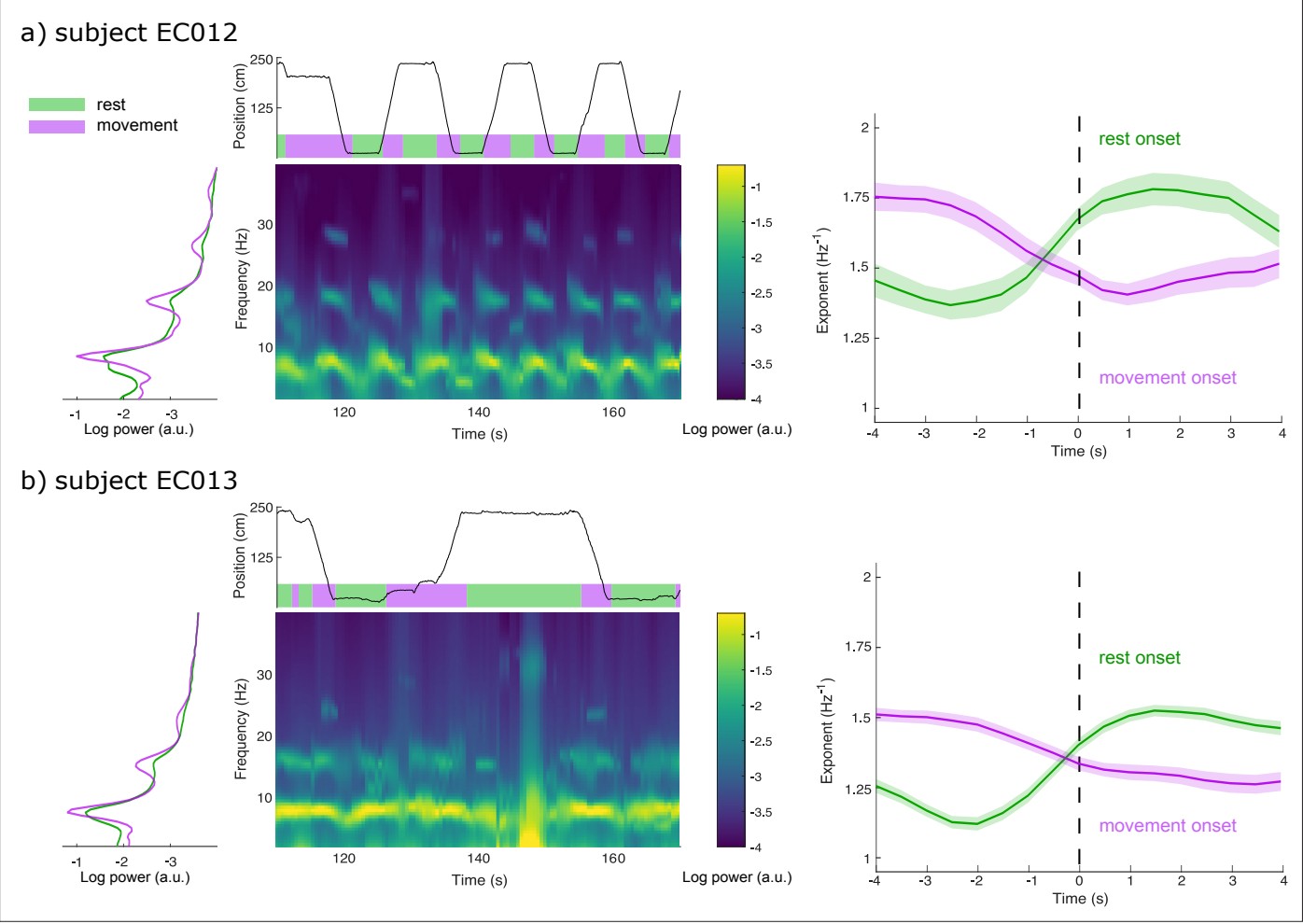

**Figure 5.** Spectral Parameterization Resolved in Time (SPRiNT) captures aperiodic dynamics related to locomotion. (**a**) We derived the data periodograms collapsed across rest (green) and movement (purple) periods for subject EC012 and observed broad increases in signal power during rest compared to movement, below 20 Hz. A representative SPRiNT spectrogram is shown. The time series of the subject's position is shown in the top plot (green: rest; purple: movement). We observed gradual shifts of aperiodic exponent around the occurrence of locomotor transitions (right plot), with increasing exponents at the onset of rest (green curve) and decreasing exponents at movement onset (purple curve). Solid lines indicated trial mean, with shaded area showing the 95% CI. (**b**) Same data as (**a**) but for subject EC013. The data samples consisted of, for EC012, 62 epochs of rest onset and 81 epochs of movement onset; for EC013, 303 epochs of rest onset and 254 epochs of movement onset.

The online version of this article includes the following source data and figure supplement(s) for figure 5:

**Source data 1.** Empirical distributions of Spectral Parameterization Resolved in Time (SPRiNT) aperiodic parameters.

**Figure supplement 1.** Examples of sawtooth rhythms from two representative electrodes in entorhinal cortex layer 3 from both subjects.

**Figure supplement 2.** Empirical distributions of Spectral Parameterization Resolved in Time (SPRiNT) aperiodic exponent and offset parameters.

**Figure supplement 3.** Temporal variability of aperiodic exponent during transitions between movement and rest is partially explained by movement speed.

Learning from the *specparam* experience, we expect that more practical (and critical) recommendations will emerge and be shared by more users adopting SPRiNT, with the pivotal expectation, as with all analytical methods in neuroscience (*Salmelin and Baillet, 2009*), that users carefully and critically review the sensibility of the outcome of SPRiNT parameterization applied to their own data and to their own neuroscience questions (*Ostlund et al., 2022*).

## Future directions

We used the STFT as the underlying time-frequency decomposition technique for SPRiNT. A major asset of STFT is computational efficiency, but with sliding time windows of fixed duration, the method

is less sophisticated that wavelet alternatives in terms of trading-off between temporal specificity and frequency resolution (*Cohen, 2014*). Combining *specparam* with STFT yields rapid extraction of spectral parameters from time-frequency data. In principle, spectral parameterization should be capable of supplementing any time-frequency decomposition technique, such as wavelet transforms (*Pietrelli et al., 2021*), though at the expense of significantly greater computational cost. However, we have shown that the wavelet-*specparam* alternative to SPRiNT underperformed to recover aperiodic signal components. Further, the temporal smoothing necessary to reduce wavelet-*specparam* parameter estimation errors to levels similar to SPRiNT's (4 s Gaussian kernel; *Figure 2*) yields substantial redundancy of the spectral parameterization following wavelet decompositions. Another alternative to using STFT would be the recent *superlet* approach (*Moca et al., 2021*), which was designed to preserve a fixed resolution across time and frequency. Combining *superlets* with *specparam* is to be explored, although reduced computational cost remains a very practical benefit of STFT.

Scientific interest towards aperiodic neurophysiological activity has recently intensified, especially in the context of methodological developments for the detection of transient oscillatory activity in electrophysiology (*Brady and Bardouille, 2022*; *Seymour et al., 2022*). These methods first remove the aperiodic component from power spectra using *specparam* before detecting oscillatory bursts from wavelet spectrograms. SPRiNT's outlier peak removal procedure also detects burst-like spectrographic components, although for a different purpose. SPRiNT is one methodological response for measuring and correcting for aperiodic spectral components and, as such, could contribute to improve tools for detecting oscillatory bursts, as suggested by *Seymour et al., 2022*.

Future ameliorations for SPRiNT to determine the parameters of periodic components (number of peaks and peak amplitude) may be driven by a model selection approach based, e.g., on the Bayesian information criterion (*Schwarz, 1978*), which would advantage models with the most parsimonious number of periodic components in the data.

In conclusion, the SPRiNT algorithm enables the parameterization of the neurophysiological spectrogram. We validated the time tracking of periodic and aperiodic spectral features with a large sample of ground-truth synthetic time series and empirical data including human resting-state and rodent intracranial electrophysiological recordings. We showed that SPRiNT provides estimates of dynamic fluctuations of aperiodic and periodic neural activity that are related to meaningful demographic or behavioural outcomes. We anticipate that SPRiNT and future related developments will augment the neuroscience toolkit and enable new advances in the characterization of complex neural dynamics.

## Methods

SPRiNT runs on individual time series and returns a parameterized representation of the spectrogram. The algorithm first derives STFTs over time windows that slide on the time series. Second, the modulus of STFT coefficients is averaged over *n* consecutive time windows to produce smoothed PSD estimates at each time bin. Third, each of the resulting PSDs is parameterized into periodic and aperiodic components, using the *specparam* algorithm. A fourth optional step consists of the removal of outlier periodic components from the raw SPRiNT spectrograms. We developed SPRiNT as a plug-in library that interoperates with *Brainstorm* (*Tadel et al., 2011*) and therefore is an open-source and accessible to everyone.

### Parameterization of short-time periodograms

STFTs are computed iteratively on sliding time windows (default window length = 1 s; tapered by a Hann window) using MATLAB's fast Fourier transform (R2020a; Natick, MA, USA). Each window overlaps with its nearest neighbours (default overlap = 50%). The modulus of Fourier coefficients of the running time window is then averaged locally with those from preceding and following time windows, with the number of time windows included in the average, *n*, determined by the user (default is n=5; *Figure 1A*). The resulting periodogram is then parameterized with *specparam*. The resulting spectrogram is time-binned based on time points located at the centre of each sliding time window.

### Tracking periodic and aperiodic components across time

We used the MATLAB implementation of *specparam* in *Brainstorm* (*Tadel et al., 2011*), adapted from the original Python code (version 1.0.0) by *Donoghue et al., 2020*. The aperiodic component of the

power spectrum is typically represented using two parameters (exponent and offset); an additional knee parameter is added when a bend is present in the aperiodic component (*Donoghue et al., 2020*; *Donoghue et al., 2020*). Periodic components are parameterized as peaks emerging from the aperiodic component using Gaussian functions controlled with three parameters (mean [centre frequency], amplitude, and SD).

For algorithmic speed optimization purposes, in each iteration of *specparam* across time, the optimization of the aperiodic exponent is initialized from its *specparam* estimate from the preceding time bin. All other parameter estimates are initialized using the same data-driven approaches as *specparam* (*Donoghue et al., 2020*).

## Pruning of periodic component outliers

We derived a procedure to remove occasional peaks of periodic activity from parameterized spectrograms and emphasize expressions of biologically plausible oscillatory components across successive time bins. This procedure removes peaks with fewer than a user-defined number of similar peaks (by centre frequency; default = 3 peaks within 2.5 Hz) within nearby time bins (default = 6 bins). This draws from observations in synthetic data that non-simulated peaks are parameterized in isolation (few similar peaks in neighbouring time bins; *Figure 1—figure supplement 1*). Aperiodic parameters are refit at time bins where peaks have been removed, and models are subsequently updated to reflect changes in parameters. This post-processing procedure is applied on all SPRiNT outputs shown but remains optional (albeit recommended).

## Study 1: Time series simulations

We simulated neural time series using in-house code based on the NeuroDSP toolbox (*Cole et al., 2019*) with MATLAB (R2020a; Natick, MA, USA). The time series combined aperiodic with periodic components (*Donoghue et al., 2020*). Each simulated 60 s time segment consisted of white noise sampled at 200 Hz generated with MATLAB's coloured noise generator (R2020a; Natick, MA, USA). The time series was then Fourier-transformed (frequency resolution = 0.017 Hz) and convolved with a composite spectrogram of simulated aperiodic and periodic dynamics (temporal resolution = 0.005 s). The final simulated time series was generated as the linear combination of cosines of each sampled frequency (with random initial phases), with amplitudes across time corresponding to the expected power from the spectrogram.

### Simulations of transient and chirping periodic components

The aperiodic exponent was initialized to 1.5 Hz$^{-1}$ and increased to 2.0 Hz$^{-1}$, and offset was initialized to −2.56 a.u. and increased to −1.41 a.u.; both linearly increasing between the 24 s and 36 s time stamps of the time series. Periodic activity in the alpha band (centre frequency = 8 Hz, amplitude = 1.2 a.u., and SD = 1.2 Hz) was generated between time stamps 8 s and 40 s, as well as between 41–46 s and 47–52 s. Periodic activity in the beta band (centre frequency = 18 Hz, amplitude = 0.9 a.u., and SD = 1.4 Hz) was generated between 15 and 25 s and down-chirped linearly from 18 to 15 Hz between 18 and 22 s. Peak amplitude was calculated as the relative height above the aperiodic component at every sampled frequency and time point. The SNR for peaks is reflected in their respective amplitudes, with peaks of lower amplitude exhibiting lower SNRs. All amplitudes of periodic activity were tapered by a Tukey kernel (cosine fraction = 0.4). Aperiodic and periodic parameters (and their dynamics) were combined to form a spectrogram of simulated activity.

All simulations (n=10,000) were unique as each was generated from a unique white-noise time series seed, and the cosine waves to simulate periodic components were each assigned a random initial phase.

Each simulated time series was analysed with SPRiNT using 5×1 s sliding time windows with 50% overlap (frequency range: 1–40 Hz). Settings for *specparam* were: peak width limits: (0.5 6); maximum number of peaks: 3; minimum peak amplitude: 0.6 a.u.; peak threshold (minimum peak SNR): 2.0 SDs; proximity threshold: 2 SDs; aperiodic mode: fixed. Settings for peak post-processing were: number of neighbouring peaks: 3; centre frequency: 2.5 Hz; time bin: 6 bins (=3 s). Periodic alpha activity was identified using the highest amplitude peak parameterized in each time bin between 5.5 and 10.5 Hz, while periodic beta activity was identified using the highest amplitude peak in each time bin between 13.5 and 20.5 Hz.

We also parameterized Morlet wavelet spectrograms of each simulated time series using *specparam* (***Donoghue et al., 2020***; MATLAB version). Wavelet transforms were computed with *Brainstorm* (***Tadel et al., 2011***; 1–40 Hz, in 1 Hz steps) using default settings (central frequency = 3 Hz, full width at half maximum [FWHM] = 1 s). Before parameterizing wavelet transforms, we applied a 4 s temporal smoothing filter (Gaussian kernel, SD = 1 s; time range: 3.5–56.5 s, in 0.005 s steps) to increase SNR (results prior to this step are shown for the first 1000 simulations in Supplemental materials). Settings for *specparam* were: peak width limits: (0.5 6); maximum number of peaks: 3; minimum peak amplitude: 0.6 a.u.; peak threshold: 2.0 SDs; proximity threshold: 2 SDs; aperiodic mode: fixed. Periodic alpha activity was identified using the highest amplitude peak parameterized in each time bin between 5.5 and 10.5 Hz. Periodic beta activity was identified using the highest amplitude peak in each time bin between 13.5 and 20.5 Hz.

Model fit error was calculated as the MAE between expected and modelled spectral power by each component across simulations and times. Algorithmic performances were assessed by calculating MAE in parameter estimates across simulations and time points relative to expected parameters. Peak-fitting probability in the alpha (5.5–10.5 Hz) and beta (13.5–20.5 Hz) bands was calculated for each time bin as the fraction of simulations with at least one oscillatory peak recovered in the frequency band of interest.

## Generic time series simulations

For each time series generation, we sampled the parameter values of their arhythmic/rhythmic components uniformly from realistic ranges. Aperiodic exponents were initialized between 0.8 and 2.2 Hz$^{-1}$. Aperiodic offsets were initialized between –8.1 and –1.5 a.u. Within the 12–36 s time segment into the simulation (onset randomized), the aperiodic exponent and offset underwent a linear shift of magnitude in the ranges –0.5–0.5 Hz$^{-1}$ and –1–1 a.u. (sampled continuously and chosen randomly), respectively. The duration of the linear shift was randomly selected for each simulated time series between 6 and 24 s. Between zero and four oscillatory (rhythmic) components were added to each trial with parameters randomly sampled within the following ranges: centre frequency: 3–35 Hz; amplitude: 0.6–1.6 a.u.; SD: 1–2 Hz. The onset (5–40 s) and duration (3–20 s) of each of the rhythmic components were also randomized across components and across trials, with the constraint that they would not overlap both in time and frequency; they were allowed to overlap in either dimension. If a rhythmic component overlapped temporally with another one, its centre frequency was set at least 2.5 peak SDs from the other temporally overlapping rhythmic component(s). The magnitude of each periodic component was tapered by a Tukey kernel (cosine fraction = 0.4).

Each simulation was analysed with SPRiNT using 5×1 s STFT windows with 50% overlap (frequency range: 1–40 Hz). Settings for *specparam* were: peak width limits: (0.5 6); maximum number of peaks: 6; minimum peak amplitude: 0.6 a.u.; peak threshold: 2.0 SDs; proximity threshold: 2 SDs; aperiodic mode: fixed. Settings for peak post-processing were: number of neighbouring peaks: 3; centre frequency: 2.5 Hz; time bin: 6 bins (=3 s). The spectrogram outcome of SPRiNT was analysed to identify rhythmic components as correct (i.e., present in ground truth signal) or incorrect components. Rhythmic SPRiNT components were labelled as correct if their centre frequency was within 2.5 peak SDs from any of the ground truth rhythmic components. In the event of multiple SPRiNT rhythmic components meeting these conditions, we selected the one with the largest amplitude peak (marking the other as incorrect).

Errors on parameter estimates were assessed via MAE measures with respect to their ground truth values. The peak-fitting probability for each simulated rhythmic component was derived as the fraction of correct peaks recovered when one was expected. Model fit error was calculated for each time bin as the MAE between empirical and SPRiNT spectral power. We used a linear regression model (MATLAB's *fitlm*; 2020a; Natick, MA, USA) to predict model fit errors at each time bin, using number of simulated peaks as a predictor.

$$MAE = intercept + B * number\ of\ simulated\ peaks$$

We also simulated 1000 time series with aperiodic activity featuring a static knee (***Figure 3—figure supplement 2***). Aperiodic exponents were initialized between 0.8 and 2.2 Hz$^{-1}$. Aperiodic offsets were initialized between –8.1 and –1.5 a.u., and knee frequencies were set between 0 and 30 Hz. Within the 12–36 s time segment into the simulated time series (onset randomized), the aperiodic exponent

and offset underwent a linear shift and a random magnitude in the range of –0.5 to 0.5 Hz$^{-1}$ and –1 to 1 a.u., respectively. The duration of the linear shift was randomly selected for each simulated time series between 1 and 20 s; the knee frequency was constant for each simulated time series. We added two oscillatory (rhythmic) components (amplitude: 0.6–1.6 a.u.; SD: 1–2 Hz) of respective peak centre frequencies between 3 and 30 Hz and between 30 and 80 Hz, with the constrain of minimum peak separation of at least 2.5 peak SDs. The onset of each periodic component was randomly assigned between 5 and 25 s, with an offset between 35 and 55 s.

We analysed each simulated time series with SPRiNT using 5×1 s STFT windows with 50% overlap over the 1–100 Hz frequency range. Parameter settings for *specparam* were: peak width limits: (0.5 6); maximum number of peaks: 3; minimum peak amplitude: 0.6 a.u.; peak threshold: 2.0 SDs; proximity threshold: 2.0 SDs; aperiodic mode: knee. Settings for peak post-processing were: number of neighbouring peaks: 3; centre frequency: 2.5 Hz; time bin: 6 bins (=3 s). The identification of periodic components was registered as correct or incorrect using the methods described above. We discarded the time bins (<2%) where aperiodic exponent estimations did not converge within the expected range.

## Study 2: Resting-state electrophysiology data

We used open-access resting-state EEG and demographics data collected for 212 participants from the LEMON (*Babayan et al., 2019*). Data from the original study was collected in accordance with the Declaration of Helsinki, and the study protocol was approved by the ethics committee at the medical faculty of the University of Leipzig (reference No. 154/13-ff). Participants were asked to alternate every 60 s between eyes-open and eyes-closed resting-state for 16 min. Continuous EEG activity (2500 Hz sampling rate) was recorded from 61 Ag/AgCl active electrodes placed in accordance with the 10–10 system. An electrode below the right eye recorded eyeblinks (ActiCap System, Brain Products). Impedance of all electrodes was maintained below 5 kΩ. EEG recordings were referenced to electrode FCz during data collection (*Babayan et al., 2019*) and re-referenced to an average reference during preprocessing.

Preprocessing was performed using *Brainstorm* (*Tadel et al., 2011*). Recordings were resampled to 250 Hz before being high-pass filtered above 0.1 Hz using a Kaiser window. Eyeblink EEG artefacts were detected and attenuated using signal-space projection. Data was visually inspected for bad channels and artefacts exceeding 200 μV. 20 participants were excluded for not following task instructions, 2 for failed EEG recordings, 1 for data missing event markers, and 11 were excluded for EEG data of poor quality (>8 bad sensors). The results herein are from the remaining 178 participants (average number of bad sensors = 3). We extracted the first 5 min of consecutive quality data, beginning with the eyes-closed condition, from electrode Oz for each participant. We removed 2.5 s of data centred at transitions between eyes-open and eyes-closed from further analyses due to sharp changes observed in aperiodic parameters when participants transitioned between eyes-open and eyes-closed (*Figure 4—figure supplement 1*) likely to be artefactual residuals of eye movements.

### Spectrogram analysis

Each recording block was analysed with SPRiNT using 5×1 s sliding time windows with 50% overlap (frequency range: 1–40 Hz). We ran SPRiNT using *Brainstorm* with the following settings: peak width limits: (1.5 6); maximum number of peaks: 3; minimum peak amplitude: 0.5 a.u.; peak threshold: 2.0 SDs; proximity threshold: 2.5 SDs; aperiodic mode: fixed. Peak post-processing was run on SPRiNT outputs number of neighbouring peaks 3; centre frequency: 2.5 Hz; time bin: 6 bins (=3 s). Alpha peaks were defined as all periodic components detected between 6 and 14 Hz. To capture variability in alpha-peak centre frequency across time, mean and SDs of alpha-peak centre frequency distributions were computed across both the eyes-open and eyes-closed conditions and by age group (defined below).

We computed spectrograms from Morlet wavelet time-frequency decompositions (1–40 Hz, in 1 Hz steps) using *Brainstorm* (with default parameters; central frequency = 1 Hz, FWHM = 3 s; *Tadel et al., 2011*). We also parameterized periodograms across eyes-open and eyes-closed time segments using *specparam* with *Brainstorm*, with the following settings: frequency range: 1–40 Hz; peak width limits: (0.5 6); maximum number of peaks: 3; minimum peak amplitude: 0.2 a.u.; peak threshold: 2.0 SDs; proximity threshold: 1.5 SDs; aperiodic mode: fixed.

## Contrast between eyes-open and eyes-closed conditions

All regression analyses were performed in R (V 3.6.3; *R Development Core Team, 2020*). We ran a logistic regression model whereby we predicted the condition (i.e., eyes-open vs eyes-closed) from the mean and SD of the following SPRiNT parameters: alpha centre frequency, alpha power, and aperiodic exponent. All model predictors were entered as fixed effects. Significance of each beta coefficient was tested against zero (i.e., $B_n = 0$). We quantified the evidence for each predictor in our models with a Bayes factor analysis where we systematically removed one of the predictors and computed the Bayes factor using the *BayesFactor* library (*Morey and Rouder, 2018*). We compared the most complex model (i.e., the full model) against all models formulated by removing a single predictor. Evidence in favour of the full model (i.e., BF <1) indicated that a given predictor improved model fit, whereas evidence for the model without the predictor (i.e., BF >1) showed limited improvement in terms of model fit.

We also fitted a logistic regression model to predict experimental condition (i.e., eyes-open and eyes-closed; dummy coded) from mean alpha-band power (6–14 Hz) entered as a fixed effect. Alpha-band power was computed as the mean log-power between 6 and 14 Hz for each condition extracted from the Morlet wavelets spectrograms. Significance of each beta coefficient was tested against zero (i.e., $B_n = 0$). Finally, we adjusted a logistic regression model to predict behavioural condition (eyes-open vs eyes-closed) from *specparam* parameters (aperiodic exponent, alpha-peak centre frequency, and alpha-peak power) as fixed effects, where significance of each beta coefficient was tested against zero (i.e., $B_n = 0$).

## Predicting age from resting-state activity

Participants were assigned to two groups based on their biological age: younger adults (age: 20–40 years, n=121) and older adults (age: 55–80 years, n=57). The SPRiNT-modelled alpha peaks and aperiodic parameters were collapsed across time to generate condition-specific distributions of model parameters per participant. We used these distributions to examine the mean and SD of alpha centre frequency, alpha power, and aperiodic exponent. We fitted two logistic regression models using the *glm* function in R (*R Development Core Team, 2020*) for the eyes-open and eyes-closed conditions:

$$
\begin{aligned}
age = \quad & intercept + B1 * mean\ alpha\ center\ frequency + B2 \\
& * standard\ deviation\ alpha\ center\ frequency + B3 * mean\ alpha\ power + B4 \\
& * standard\ deviation\ alpha\ power + B5 * mean\ aperiodic\ slope + B6 \\
& * standard\ deviation\ aperiodic\ slope \\
age = \quad & intercept + B1 * mean\ alpha\ center\ frequency + B2 \\
& * standard\ deviation\ alpha\ center\ frequency + B3 * mean\ alpha\ power + B4 \\
& * standard\ deviation\ alpha\ power + B5 * mean\ aperiodic\ slope + B6 \\
& * standard\ deviation\ aperiodic\ slope
\end{aligned}
$$

All predictors were entered as fixed effects. Significance of each beta coefficient was tested against zero (i.e., $B_n = 0$). We also quantified the evidence for each predictor in our models with a Bayes factor analysis. We performed similar logistic regressions using data from Morlet wavelets spectrograms and *specparam*-modelled power spectra (using the same parameters as those used for predicting behavioural condition). Finally, we performed logistic regressions using the mean and temporal variability (SD) of individual alpha-peak frequency (the frequency corresponding to the maximum power value between 6 and 14 Hz; *Klimesch, 1999*) derived from the STFT in both conditions to predict age group.

## Study 3: Intracranial rodent data

Local field potential (LFP) recordings and animal behaviour, originally published by *Mizuseki et al., 2009*, were collected from two Long-Evans rats (data retrieved from https://crcns.org). Animals were implanted with eight-shank multi-site silicon probes (200 µm inter-shank distance) spanning multiple layers of dorsocaudal medial entorhinal cortex (entorhinal cortex, dentate gyrus, and hippocampus). Neurophysiological signals were recorded while animals traversed to alternating ends of an elevated

linear track (250 × 7 cm) for 30 µL water reward (animals were water deprived for 24 hr prior to task). All surgical and behavioural procedures in the original study were approved by the Institutional Animal Care and Use Committee of Rutgers University (protocol No. 90–042). Recordings were acquired continuously at 20 kHz (RC Electronics) and bandpass-filtered (1 Hz-5 kHz) before being down-sampled to 1250 Hz. In two rats (EC012 and EC013), nine recording blocks of activity in entorhinal cortex layer 3 (EC3) were selected for further analysis (16 electrodes in EC012 and 8 electrodes in EC013). Electrodes in EC012 with consistent isolated signal artefacts were removed (average number of bad electrodes = 2; none in EC013). Movement-related artefacts (large transient changes in LFP across all electrodes, either positive or negative) were identified by visual inspection and data coinciding with these artefacts were later discarded from further analysis. Animal head position was extracted from video recordings (39.06 Hz) of two head-mounted LEDs and temporally interpolated to align with SPRiNT parameters across time (piecewise cubic Hermite interpolative polynomial; MATLAB's *pchip*; 2020a; Natick, MA, USA).

## Spectrogram analysis

Each recording block was analysed with SPRiNT using 5×2 s sliding time windows with 75% overlap (frequency range: 2–40 Hz). The 1 Hz frequency bin was omitted from spectral analyses due to its partial attenuation by the bandpass filter applied to the data. Time windows of 2 s were used to increase frequency resolution, with an overlap ratio of 75% to preserve the temporal resolution of 0.5 s and to increase the temporal specificity of the spectrogram windows. Settings for *specparam* were set: peak width limits: (1.5 5); maximum number of peaks: 3; minimum peak amplitude: 0.5 a.u.; peak threshold: 2.0 SDs; proximity threshold: 2.0 SDs; aperiodic mode: fixed. Settings for peak post-processing were set as: number of neighbouring peaks: 3; centre frequency bounds: 2.5 Hz; time bin bounds: 6 bins (=3 s). Aperiodic parameters were averaged across electrodes and aligned with behavioural data.

## Tracking aperiodic dynamics during movement transitions

Time bins were categorized based on whether animals were resting at either end of the track or moving towards opposite ends of the track ('rest' or 'movement', respectively) using animal position (and speed). Rest-to-movement and movement-to-rest transitions were defined as at least four consecutive seconds of rest followed by four consecutive seconds of run (t=0 s representing the onset of movement) or vice versa (t=0 s representing the onset of rest), respectively. In both subjects, we also fit separate linear regression models (MATLAB's *fitlm*; 2020a; Natick, MA, USA) of the relation between aperiodic exponents and movement speed at the transitions between movement and rest.

## Software and code availability

The SPRiNT algorithm and all code needed to produce the figures shown are available from GitHub (https://github.com/lucwilson/SPRiNT; *Wilson, 2022*). The SPRiNT algorithm is also available from the *Brainstorm* distribution (*Tadel et al., 2011*).

## Acknowledgements

L.E.W. acknowledges the support of an NSERC Undergraduate Student Research Award. J.D.S.C. acknowledges the support of the Alexander Graham-Bell Doctoral NSERC fellowship. S.B. is grateful for the support received from the NIH (R01 EB026299), a Discovery Grant from the Natural Science and Engineering Research Council of Canada (436355–13), the CIHR Canada Research Chair in Neural Dynamics of Brain Systems, the Brain Canada Foundation with support from Health Canada, and the Innovative Ideas program from the Canada First Research Excellence Fund, awarded to McGill University for the Healthy Brains for Healthy Lives initiative. This research was undertaken thanks in part to funding from the Canada First Research Excellence Fund, awarded to McGill University for the Healthy Brains for Healthy Lives initiative.

## Additional information

### Funding

| Funder | Grant reference number | Author |
|---|---|---|
| Natural Sciences and Engineering Research Council of Canada | Undergraduate Student Research Award | Luc Edward Wilson |
| Natural Sciences and Engineering Research Council of Canada | Alexander Graham-Bell NSERC Doctoral Fellowship | Jason da Silva Castanheira |
| National Institutes of Health | R01 EB026299 | Sylvain Baillet |
| Natural Sciences and Engineering Research Council of Canada | Discovery Grant,436355-13 | Sylvain Baillet |
| Canadian Institutes of Health Research | Canada Research Chair in Neural Dynamics of Brain Systems | Sylvain Baillet |
| Health Canada | Brain Canada Foundation | Sylvain Baillet |
| Canada First Research Excellence Fund | Innovative Ideas program | Sylvain Baillet |

The funders had no role in study design, data collection and interpretation, or the decision to submit the work for publication.

### Author contributions

Luc Edward Wilson, Jason da Silva Castanheira, Conceptualization, Data curation, Software, Formal analysis, Visualization, Methodology, Writing – original draft, Project administration, Writing – review and editing; Sylvain Baillet, Conceptualization, Supervision, Funding acquisition, Writing – original draft, Project administration, Writing – review and editing

### Author ORCIDs

Luc Edward Wilson http://orcid.org/0000-0002-9431-9527
Jason da Silva Castanheira http://orcid.org/0000-0002-7022-9009
Sylvain Baillet http://orcid.org/0000-0002-6762-5713

### Ethics

Human subjects: Data from the MPI-Leipzig Mind-Brain-Body dataset was collected in accordance with the Declaration of Helsinki and the study protocol was approved by the ethics committee at the medical faculty of the University of Leipzig (reference No. 154/13-ff).
All surgical and behavioural procedures in the HC3 study were approved by the Institutional Animal Care and Use Committee of Rutgers University (protocol No. 90-042). All surgery was performed under isoflurane anesthesia.

### Decision letter and Author response

Decision letter https://doi.org/10.7554/eLife.77348.sa1
Author response https://doi.org/10.7554/eLife.77348.sa2

## Additional files

### Supplementary files

• Transparent reporting form

### Data availability

The SPRiNT algorithm and all code needed to produce the figures shown are available from GitHub (https://github.com/lucwilson/SPRiNT, copy archived at swh:1:rev:ba6820f010ed80e20ffe-502562ab55f515d98e3f). The simulated data (as well as source data for Figure 2) are openly available

on OSF (https://doi.org/10.17605/OSF.IO/UGZJA). Resting-state EEG data was obtained from the open repository LEMON (https://openneuro.org/datasets/ds000221/versions/00002). Intracranial rodent data (study HC3) is openly available from Mizuseki et al. 2009 (https://crcns.org). Figure 3 - source data 1, Figure 4 - source data 1, and Figure 5 - source data 1 contain the numerical data used to generate the figures.

The following datasets were generated:

| Author(s) | Year | Dataset title | Dataset URL | Database and Identifier |
|---|---|---|---|---|
| Wilson L, da Silva Castanheira J | 2022 | SPRiNT: Time-resolved parameterization of aperiodic and periodic brain activity | https://doi.org/10.5281/zenodo.6903693 | Zenodo, 10.5281/zenodo.6903693 |
| Wilson L, da Silva Castanheira J | 2022 | Time-resolved parameterization of aperiodic and periodic brain activity | https://doi.org/10.17605/OSF.IO/UGZJA | Open Science Framework, 10.17605/OSF.IO/UGZJA |

The following previously published datasets were used:

| Author(s) | Year | Dataset title | Dataset URL | Database and Identifier |
|---|---|---|---|---|
| Babayan A, Baczkowski B, Cozatl R, Dreyer M, Engen H, Erbey M, Falkiewicz M, Farrugia N, Gaebler M, Golchert J, Golz L, Gorgolewski K, Haueis P, Huntenburg J, Jost R, Kramarenko Y, Krause S, Kumral D, Lauckner M, Margulies DS, Mendes N, Ohrnberger K, Oligschläger S, Osoianu A, Pool J, Reichelt J, Reiter A, Röbbig J, Schaare L, Smallwood J, Villringer A | 2018 | The MPI-Leipzig Mind-Brain-Body dataset | https://openneuro.org/datasets/ds000221/versions/00002 | OpenNeuro, 10.18112/openneuro.ds000221.v1.0.0 |
| Mizuseki K, Sirota A, Pastalkova E, Diba K, Buzsáki G | 2013 | Multiple single unit recordings from different rat hippocampal and entorhinal regions while the animals were performing multiple behavioral tasks | http://doi.org/10.6080/K09G5JRZ | Collaborative Research in Computational Neuroscience, 10.6080/K09G5JRZ |

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

# Appendix 1

## Supplemental materials

### Detection and removal of spectrogram outlier components

A known issue of specparam is the fitting of spurious, outlier spectral peaks (*Donoghue et al., 2020*). With Spectral Parameterization Resolved in Time (SPRiNT), these peaks often appear as transient episodes of periodic activity in the spectrogram. We propose a post-processing option in SPRiNT to detect and remove fast, transient periodic activity (*Figure 1—figure supplement 1*). In short, the procedure searches for clusters of spectral peaks over a user-defined maximum time period (see Methods "Pruning of periodic component outliers"). Once an outlier peak is detected at a given time bin and removed from the model, aperiodic parameters are refit with specparam to account for the variance previously attributed to the spurious peak, and models are subsequently updated to reflect changes in parameters.

### Wavelet-specparam with alternative wavelet parameters (synthetic data challenge I)

In addition to the wavelet settings used in the main text, we parameterized Morlet wavelet spectrograms of the first 1000 simulated time series from challenge I using alternative full width at half maximum (FWHM) settings for the wavelet transforms. This resulted in lower FWHM yielding wavelet spectrograms of higher temporal and lower spectral resolution (*Cohen, 2014*). As with other analyses of this dataset, settings for *specparam* were: peak width limits: (0.5 6); maximum number of peaks: 3; minimum peak amplitude: 0.6 a.u.; peak threshold: 2.0 SDs; proximity threshold: 2.0 SDs; aperiodic mode: fixed.

Wavelet settings of finer resolution in time and coarser in frequency (time range: 3–57 s, in 0.005 s steps; central frequency = 1 Hz, FWHM = 2 s; frequency range: 1–40 Hz, in 1 Hz steps) yielded lower estimation errors of exponent (mean absolute error [MAE] = 0.12) and offset (MAE = 0.35) compared to original settings (exponent and offset MAE = 0.19 and 0.78). Alpha peaks were recovered with higher sensitivity (97% at time bins with maximum peak amplitude and original 95%) and specificity (32% spurious detections and original 47%), although with greater errors in centre frequency (MAE = 0.61), amplitude (MAE = 0.25), and bandwidth (MAE = 0.94) compared to original settings (centre frequency, amplitude, and bandwidth MAE = 0.41, 0.24, and 0.64, respectively). Down-chirping beta oscillations were detected with lower sensitivity (29% sensitivity at time bins with maximum peak amplitude and original 62%) but higher specificity (97%, original 90%), and with greater errors in centre frequency (MAE = 0.63), amplitude (MAE = 0.17), and bandwidth (MAE = 1.59) relative to original settings (centre frequency, amplitude, and bandwidth MAE = 0.58, 0.16, and 1.05, respectively).

When wavelet settings prioritized resolution in frequency over time (time range: 4–56 s, in 0.005 s steps; central frequency = 1 Hz, FWHM = 4 s; frequency range: 1–40 Hz, in 1 Hz steps) relative to original settings, the errors in estimates of exponent (MAE = 0.16) and offset (MAE = 0.47) parameters were reduced (original exponent and offset MAE = 0.19 and 0.78, respectively). Alpha peaks were recovered with higher sensitivity (99% at time bins with maximum peak amplitude and original 95%) and similar specificity (46% spurious detections and original 47%), although with larger errors in centre frequency (MAE = 0.33), amplitude (MAE = 0.20), and bandwidth (MAE = 0.43) compared to original settings (centre frequency, amplitude, and bandwidth MAE = 0.41, 0.24, and 0.64, respectively). In contrast, down-chirping beta oscillations were detected with slightly higher sensitivity (79% at time bins with maximum peak amplitude and original 62%) and specificity (91%, original 90%), and with lower errors on centre frequency (MAE = 0.37), amplitude (MAE = 0.14), and bandwidth (MAE = 0.71) compared to original settings (centre frequency, amplitude, and bandwidth MAE = 0.58, 0.16, and 1.05, respectively).

### SPRiNT with alternative short-time Fourier transform (STFT) parameters (synthetic data challenge I)

In addition to the primary SPRiNT settings used in the main text (i.e., 5×1 s windows with 50% overlap), we parameterized STFT spectrograms of the first 1000 simulated time series from challenge I using alternative settings for the STFTs (*Figure 2—figure supplement 3*). One setting enabled higher temporal resolution (5×1 s with 75% overlap), while the other enabled higher frequency

resolution (5×2 s with 75% overlap). As with other analyses of this dataset, settings for *specparam* were: peak width limits: (0.5 6); maximum number of peaks: 3; minimum peak amplitude: 0.6 a.u.; peak threshold: 2.0 SDs; proximity threshold: 2.0 SDs; aperiodic mode: fixed.

SPRiNT settings for higher temporal resolution (time range: 1–59 s, in 0.25 s steps; frequency range: 1–40 Hz, in 1 Hz steps) provided slightly larger estimation errors of exponent (MAE = 0.15) and offset (MAE = 0.20) relative to original settings (exponent and offset MAE = 0.11 and 0.14, respectively). Alpha peaks were recovered with slightly lower sensitivity (98% at time bins with maximum peak amplitude; original 99%) and specificity (9% spurious detections; original 4%), and with greater errors in centre frequency (MAE = 0.43), amplitude (MAE = 0.24), and bandwidth (MAE = 0.53) compared to original settings (centre frequency, amplitude, and bandwidth MAE = 0.33, 0.20, and 0.42, respectively). Down-chirping beta oscillations were detected with lower sensitivity (93% sensitivity at time bins with maximum peak amplitude, original 98%; 86% specificity, original 98%), and with greater errors in centre frequency (MAE = 0.57), amplitude (MAE = 0.22), and bandwidth (MAE = 0.57) compared to original settings (centre frequency, amplitude, and bandwidth MAE = 0.43, 0.17, and 0.48, respectively).

SPRiNT settings for higher frequency resolution (time range: 2–58 s, in 0.5 s steps; frequency range: 1–40 Hz, in 0.5 Hz steps) provided comparable estimation errors of exponent (MAE = 0.13) and offset (MAE = 0.16) relative to original settings (exponent and offset MAE = 0.11 and 0.20, respectively). Alpha peaks were recovered with similar sensitivity (99% at time bins with maximum peak amplitude; original 99%) but lower specificity (21% spurious detections; original 4%), and with comparable errors in centre frequency (MAE = 0.35), amplitude (MAE = 0.23), and bandwidth (MAE = 0.41) to original settings (centre frequency, amplitude, and bandwidth MAE = 0.33, 0.20, and 0.42, respectively). Down-chirping beta oscillations were detected with comparable sensitivity (99% sensitivity at time bins with maximum peak amplitude and original 98%) but lower specificity (78%, original 98%), and with greater errors in centre frequency (MAE = 0.50), amplitude (MAE = 0.21), and bandwidth (MAE = 0.59) relative to original settings (centre frequency, amplitude, and bandwidth MAE = 0.43, 0.17, and 0.48, respectively).

### Wavelet-*specparam* without temporal smoothing (synthetic data challenge I)

We parameterized Morlet wavelet spectrograms (central frequency = 1 Hz, FWHM = 3 s; 1–40 Hz, in 1 Hz steps) of the first 1000 simulated time series consisting of transient alpha and down-chirping beta periodic activity (time range: 1.5–58.5 s, in 0.005 s steps). In the main text, we discuss results from temporally smoothed wavelet spectrograms (see Methods). Here, we show results without temporal smoothing (*Figure 2—figure supplement 4*).

Error in estimates from unsmoothed parameterized wavelet spectrograms of exponent (MAE = 0.41) and offset (MAE = 0.83) parameters was worse than those obtained from smoothed wavelet decompositions (exponent MAE = 0.19; offset MAE = 0.78). Alpha peaks were recovered with lower sensitivity (84% at time bins with maximum peak amplitude) and specificity (41% spurious detections), and with greater errors on centre frequency (MAE = 0.82), amplitude (MAE = 0.53), and bandwidth (MAE = 0.91). The down-chirping beta oscillation was also detected with lower sensitivity (53% at time bins with maximum peak amplitude) and specificity (74%), and with greater errors on centre frequency (MAE = 1.23), amplitude (MAE = 0.60), and bandwidth (MAE = 1.10).

### SPRiNT without outlier peak removal (synthetic data challenge 1)

Here, we present the results of SPRiNT from synthetic data challenge I without outlier peak removal (*Figure 2—figure supplement 4*). For the aperiodic component, SPRiNT accurately recovered both ground truth exponent (MAE = 0.11) and offset (MAE = 0.15). It also detected the occurrences of alpha peaks with high sensitivity (99% at time bins with maximum peak amplitude) and specificity (6% spurious detections), and with low errors on their centre frequency (MAE = 0.33) and amplitude (MAE = 0.20) parameters, but overestimated the width of the periodic peak components (MAE = 0.42). SPRiNT detected and tracked the down-chirping beta periodic components with high sensitivity (95% at time bins with maximum peak amplitude) but lower specificity (95%) than with outlier peak removal (98%). Time-collapsed errors on centre frequency (MAE = 0.44) and amplitude (MAE = 0.17) parameters were low, with a tendency to overestimate the width in frequency of the periodic component (MAE = 0.48). Results following outlier peak removal are presented and discussed in the main text.

## Generalization of SPRiNT across generic aperiodic and periodic fluctuations without outlier removal (synthetic data challenge II)

We present the results of the second synthetic data challenge without outlier peak removal (*Figure 3—figure supplement 3*). SPRiNT recovered 70% of the simulated periodic components, with 73% specificity. Dynamic aperiodic exponents were recovered with a MAE of 0.13, while dynamic offsets were recovered with a MAE of 0.16. Centre frequency, amplitude, and SD parameters were recovered with MAEs of 0.46, 0.23, and 0.49, respectively.

## SPRiNT model fit error does not affect condition associations

We performed t-tests of model fit errors (MAE) between conditions and age groups. While there were no age-related effects on model fit error (eyes-open: p=0.09; eyes closed: p=0.69), we observed slightly lower model fit errors in the eyes-open condition (mean = 0.032) compared to the eyes-closed condition (mean = 0.033; $t[354] = -3.17$, p=0.002, and 95% CI [$3.0 \times 10^{-4}$ $1.3 \times 10^{-3}$]). The size of this effect was small-to-medium (Cohen's $d$=0.34).

To determine whether model fit error would affect our SPRiNT logistic regression model for condition, we included it as a fixed effect in a new logistic regression model (*Table 3*). Here, we observed the same effects for predicting condition as the original model (mean aperiodic exponent, mean alpha power, and variability of alpha power; see *Table 2*), with no significant effect of model fit error (p=0.45).

