## [Editor Report]

The paper addresses the highly timely question of how to quantify aperiodic and periodic neural activity. This was done by extending previous work by embracing time-resolved parametrization of both simulated, noninvasive EEG and intracranial data. The new approach is termed Spectral Parametrization Resolved in Time (SPRiNT) and the paper shows that the slope of aperiodic activity is linked with both behavior and age. The method thus demonstrates the importance of evaluating the state-dependence of aperiodic activity and dynamic properties of oscillatory components in a time-resolved manner, and we believe that this approach would be of great interest to researchers analyzing human electrophysiological data to address clinical and cognitive neuroscience questions.

---

## [Decision Letter]

**Decision letter after peer review:**

Thank you for submitting your article "Time-resolved parameterization of aperiodic and periodic brain activity" for consideration by *eLife*. Your article has been reviewed by 3 peer reviewers, and the evaluation has been overseen by a Reviewing Editor and Floris de Lange as the Senior Editor. The following individuals involved in the review of your submission have agreed to reveal their identity: Mats W.J. van Es (Reviewer #2); Jan Kujala (Reviewer #3).

Essential revisions:

The three reviewers were in general positive about the manuscript. However, several concerns were raised. Several of those were related to the simulated data used for testing the algorithm. Also, the comparison to other methods could be improved. Please see below for details.

*Reviewer #1 (Recommendations for the authors):*

One aspect that can improve the impact of the paper is to show how the application of SPRiNT cardinally changes the interpretation of the data in the case when it was analyzed with rather conventional approaches. For instance, one can show that while using conventional methods, there are changes in the power, yet, when SPRiNT is applied these changes disappear or vice versa. It can be either dataset presented in this study or other datasets.

Please explain more carefully how the data was generated with respect to SNR of oscillations, how SNR was explicitly controlled.

The authors write: "Each simulation was analyzed with SPRiNT using 5x1s STFT windows with 50% overlap (frequency range: 1-40 Hz)." What would happen if the fluctuation of periodic and aperiodic components occur in < 1-sec time range? This would correspond to a situation in real brain activity since E and I inputs are often transient in nature and can span just a few hundred milliseconds.

When differentiating eyes-open or eyes-closed condition, was this differentiation based on 5 mins data or on shorter segments, like a few seconds only? The latter case would be most interesting.

What are the smallest segments and number of overlapped segments for the estimation of periodic and aperiodic components? It seems that EEG and LFP data were analyzed with different parameters.

In contrast to EEG data, intracranial animal data was analyzed with different parameters for SPRiNT, i.e. each recording block was analyzed with SPRiNT using 5x2 s sliding time windows with 75% overlap. Why was it the case? For the reader, it is important to know how these decisions are made about the length of the window and overlap.

When tracking movement transitions it is important to take into account movement-related artifacts which can introduce changes in a wide frequency range. How were they handled?

The authors write: "We noted that both methods tended to overestimate peak bandwidths (Figure 2 —figure supplement 1)" It seems that there is a systematic bias in peak bandwidth estimation. Is there a way to compensate for it?

It is possible that the changes between young and old participants (or in eyes-open and eyes-closed conditions) were due to relatively local changes in low-frequency oscillations which would consequently lead to deviation from 1/f decay of spectrum. This in turn would lead to changes in goodness-of-fit (GOF) of 1/f component between conditions. Have authors observed systematic changes in GOFs between different conditions?

Please discuss cases where the aperiodic part can be stable vs when it can be unstable.

*Reviewer #2 (Recommendations for the authors):*

1. In the first analysis, the authors compare SPRiNT with specparam applied to a wavelet time-frequency spectrum. Given that the original method by Donoghue et al., is based on Welch's method (which uses the Discrete Fourier Transform; DFT), it is unclear why the authors chose wavelets as a benchmark. A more direct comparison with Donoghue et al., would be comparing the time-resolved with the static specparam approach, without the implicit comparison of STFT with wavelets. Could the authors please motivate their choice of benchmark? (potentially the analysis described in lines 36-46 of the supplement could be moved/referred to in the main text).

2. One issue when applying SPRiNT to task data is that it temporally smooths a/periodic parameter estimates (i.e. by averaging Fourier coefficients over neighbouring windows), which can lead to blurring of baseline and task windows (especially in a typical task where baseline and task period are only a few seconds). Could the authors elaborate on how to choose parameter settings (e.g., number of overlapping time windows; whether or not to fit a knee) and what pitfalls to look out for?

3. Line 194 refers to a supplemental figure (the figure 2 equivalent prior to removal of outlier peaks). Line 21 of the supplement also refers to this figure. However, the figure appears to be absent.

4. Figure 3D shows there is a general underestimation of the number of periodic components, especially in the δ band. Perhaps it would be useful to add a figure to the supplement containing a confusion matrix, i.e. showing how likely each simulated peak is to be recognised as a peak in a different frequency band (or not recognised) over all simulations.

5. Figure 3C shows the detection probability of spectral peaks with respect to centre frequency and peak amplitude. Could to authors create a similar figure for bandwidth, or at least comment on this?

6. Alternatives to the Short Time Fourier Transform (STFT) are discussed in both the introduction and discussion but do not mention Empirical Mode Decomposition (EMD; Huang et al., 1998; Quinn et al., 2021).

7. Recently, another adaptation of specparam, called PAPTO, was described by Brady and Bardouille (2022; https://doi.org/10.1016/j.neuroimage.2022.118974), specifically regarding transient oscillations. It would be good if the authors could add this in their discussion, especially in light of pruning the periodic component outliers.

8. A few observations from the Results are missing in the discussion. I would like to ask the authors to add a discussion on (1) the overestimation of peak bandwidth by SPRiNT, (2) the underestimation of the number of detected peaks in figure 3, and (3) the peaks in the aperiodic component and offset at moments of switching eyes open/closed in figure 4 – supplement 1 (also related to point 2).

9. From lines 245-248 it does not become clear from the main text that the authors conducted a logistic regression analysis; this only becomes clear from the subtext of figure 4B. Please add it to the main text.

10. It seems that tables 5 and 6 are mixed up in the main text (line 304, 309), e.g. line 304 is referring to table 5 but should be referring to table 6.

11. There appears to be a grammatical error in lines 416-418 ("…has associated how locomotor behavior is associated…").

12. Figure 3D would be clearer with a legend. Also, the subtext talks about "blue" for 3-8 Hz peaks but is ambiguous because (dark) blue also denotes an undetected peak. Please clarify in the text.

13. The reference to figure 3E in the subtext should be in bold font.

---

## [Author Response]

Reviewer #1 (Recommendations for the authors):One aspect that can improve the impact of the paper is to show how the application of SPRiNT cardinally changes the interpretation of the data in the case when it was analyzed with rather conventional approaches. For instance, one can show that while using conventional methods, there are changes in the power, yet, when SPRiNT is applied these changes disappear or vice versa. It can be either dataset presented in this study or other datasets.

We thank the Reviewer for this suggestion and agree with the significance of the proposed experiment.

We have therefore re-analyzed the LEMON dataset to provide new results obtained from the periodogram decompositions of the EEG data using specparam, in the eyes-open and eyes-closed conditions. Our goal was to evaluate whether static periodogram parameters from specparam would show effects compatible with those observed with the spectrogram parametrization derived from SPRiNT. When averaged across time, the SPRiNT α-peak amplitude in the eyes-closed condition was not predictive of age group, contrarily to the specparam α-peak amplitude in the same condition (Figure 4b). Additionally, specparam α centre frequency in the eyes-open condition was predictive of age, while SPRiNT found that the variability (but not the mean) of eyes-open α centre frequency was predictive of age, instead.

The results relating to α-peak amplitude from specparam are consistent with age-related differences reported by Donoghue et al., (2020) in the seminal specparam publication. Indeed, the detection of low-amplitude peaks is more challenging in spectrograms than in periodograms because of lesser signal averaging yielding lower signal-to-noise ratios. Further, we observed in a fraction (<10%) of LEMON participants substantial disparities between the respective estimates of eyes-closed α-peak amplitude from SPRiNT and specparam. In these participants, the spectrograms show that the presence of an α peak was sporadic (see an illustrative case in Figure 4c: the α peak is visible only during 40% of the eyes-closed condition). The fluctuations in peak presence/absence over the duration of the recording cannot be detected via a periodogram and specparam. We can therefore conclude that in this situation, α-peak amplitude in the eyes-closed condition reported by specparam may be confounded in small part by the pronounced temporal variability of the data in a subset of participants (Donoghue et al., 2021). This Discussed point is now added to the revised manuscript:

“Using specparam, we found lower α-band peak amplitudes in older individuals in the eyes-closed condition. We could not replicate this effect from spectrograms parametrized with SPRiNT (Figure 4b). This apparent divergence may be due to the challenge of detecting low-amplitude peaks in the spectrogram of older individuals. Periodograms are derived from averaging across time windows, which augments signal-to-noise ratios, and therefore the sensitivity of specparam to periodic components of lower amplitude. In a subset of participants (<10%), we also observed considerable differences between the α peak amplitudes extracted from specparam and SPRiNT, which we explained by unstable expressions of α activity over time in these participants (Figure 4C). The average α peak amplitude estimated with SPRiNT is based only on time segments when an α-band periodic component is detected. With specparam, this estimate is derived across all time windows, regardless of the presence/absence of a bona fide α component at certain time instances. The consequence is that the estimate of the average α peak amplitude is larger with SPRiNT than with specparam in these participants. Therefore, differences in α power between SPRiNT and specparam may be explained, at least in some participants, by differential temporal fluctuations of α band activity (Donoghue et al., 2021). This effect is reminiscent of recent observations that β-band power suppression during motor execution is due to sparser bursting activity, not to a sustained decrease of β-band activity (Sherman et al., 2016).” (Lines 521 to 538)

Please explain more carefully how the data was generated with respect to SNR of oscillations, how SNR was explicitly controlled.

Since our analyses focus on spectral components, we define signal-to-noise ratio for the spectrum. We defined the spectral signal-to-noise ratio in log-space (log-SNR) as the amplitude of the simulated peak component in the log-transformed periodogram divided by the standard deviation of the intrinsic spectral noise (i.e., variance unaccounted for by periodic and aperiodic components). We used 1-s time windows in all simulations, with periodograms derived from averaging Fourier coefficients over 5 overlapping windows, therefore the standard deviation of spectral noise was fixed across simulations (standard deviation = 0.2 a.u.). Hence, the log-SNR of oscillations in our simulations is equal to the simulated peak amplitude divided by 0.2 a.u. To illustrate this aspect, we provide two example spectra derived from SPRiNT (5x1-s windows with 50% overlap) with oscillatory peaks in the α band as well as their simulated peak amplitudes and derived SNRs. See Author response image 1.

**Author response image 1. sa2fig1:** Example periodic components with differing spectral signal-to-noise ratios**.** (a) Example parameterized periodogram (exponent: 1 a.u. Hz^-1^; offset: -6 a.u.; α peak centre frequency, amplitude, standard deviation: 10 Hz, 1.0 a.u., 1 Hz, respectively) with a derived log-SNR of 5. (b) Example parameterized periodogram (exponent: 1 a.u. Hz^-1^; offset: -6 a.u.; α peak centre frequency, amplitude, standard deviation: 10 Hz, 1.0 a.u., 1 Hz, respectively) with a derived log-SNR of 5. The log-SNR of each periodic component is calculated as the ratio of peak amplitude divided by one standard deviation of the intrinsic spectral noise (0.2 a.u.).

Please note that SPRiNT features an option to threshold the detection of periodic components based on their log-SNR (*peak threshold* = 2 s.d.), which can be bypassed by setting *minimum peak height* above this value (corresponding to a value greater that 0.4 a.u. for the datasets in this study). Please also note that as the number of averaged windows increases (i.e., as signal duration increases), spectral noise decreases (assuming the signal is stationary) which increases log-SNR (not shown).

The authors write: "Each simulation was analyzed with SPRiNT using 5x1s STFT windows with 50% overlap (frequency range: 1-40 Hz).” What would happen if the fluctuation of periodic and aperiodic components occur in < 1-sec time range? This would correspond to a situation in real brain activity since E and I inputs are often transient in nature and can span just a few hundred milliseconds.

Thank you for bringing up this point.

If the spectral physiological components are evolving at faster time scales than SPRiNT’s sampling rate defined from overlapping time windows, the sensitivity of the resulting spectrogram and its parametrization will be reduced. The settings used in the experiments reported (two overlapping time windows per second) would not be optimal to track phenomena at sub-second timescales. We would therefore recommend specific parameter settings to achieve higher temporal resolution: averaging Fourier coefficients over five one-second windows with 75% overlap, for instance, would provide four samples per second but with reduced spectral SNR (Figure 2 —figure supplement 3). Following the Reviewer’s suggestions (comments 3 and 5), as well as suggestion from Reviewer 2 (comments 2 and 4), we now provide practical recommendations for tuning SPRiNT parameters depending on the neuroscience question and study objectives in the revised manuscript. We also encourage the reader to consult the recommendations for *specparam* settings issued by Ostlund et al., (2021) and Gerster et al., (2022).

When differentiating eyes-open or eyes-closed condition, was this differentiation based on 5 mins data or on shorter segments, like a few seconds only? The latter case would be most interesting.

We used the entire amount of data available from the LEMON study, and namely 3 minutes in the eyes-closed condition and 2 minutes in the eyes-open condition, per participant.

The stability of spectral estimates over time, under steady behavioral conditions, has been shown to develop over 30 to 120 s of signal (Wiesman et al., 2021). We also observed sporadic changes of α activity in a subset of participants (Figure 4c). Although we sample the spectrum at sub-second resolution, we derive measures of mean and dispersion by sampling over the entire time series. The minimum recording length required for stability in measures of mean and dispersion across spectral parameters is a relevant question, but we believe that it falls outside the scope of the present study.

What are the smallest segments and number of overlapped segments for the estimation of periodic and aperiodic components? It seems that EEG and LFP data were analyzed with different parameters.In contrast to EEG data, intracranial animal data was analyzed with different parameters for SPRiNT, i.e. each recording block was analyzed with SPRiNT using 5x2 s sliding time windows with 75% overlap. Why was it the case? For the reader, it is important to know how these decisions are made about the length of the window and overlap.

We thank the Reviewer for giving us the opportunity to clarify these important aspects further.

The choice of SPRiNT time window parameters represents a trade-off between temporal resolution against spectral signal-to-noise ratio and frequency resolution.

For the intracranial data experiment, the dataset available had been high-pass filtered above 1 Hz, partially attenuating the 1-Hz frequency power bin, which motivated its removal from the parameterization process. Accordingly, we chose to increase the frequency resolution by doubling the window length to include more samples from low frequencies and better resolve the shape of the low-frequency power spectrum. To maintain a 0.5-s temporal sampling rate of the SPRiNT model, we increased the overlap ratio parameter to 75%.

As mentioned in the response to this Reviewer’s point #3 above, we now provide practical recommendations for parameter setting in the revised Discussion section. In Methods, we now explain how the SPRiNT parameters were set to analyze the intracranial data.

When tracking movement transitions it is important to take into account movement-related artifacts which can introduce changes in a wide frequency range. How were they handled?

We thank the Reviewer for pointing at possible sources of artifacts in the data that would bias result interpretation.

Originally, we removed one electrode from EC012 due to poor signal quality. However, pursuant to this Reviewer’s comment, we returned to the data to identify possible remaining movement-related artifacts. As a result, we visually identified and removed electrodes with isolated movement artifacts (only observed in EC012, mean = 2 channels removed), as well as artifacts contaminating all electrodes over <5% of all time points across both subjects. We have updated the results reported accordingly. Although all reported effects remain unaltered, we regret not noticing and removing these artifacts initially. We have edited the Methods section to detail the procedures applied for artifact detection and removal:

“Electrodes in EC012 with consistent isolated signal artifacts were removed (average number of bad electrodes = 2; none in EC013). Movement-related artifacts (large transient changes in local field potential across all electrodes, either positive or negative) were identified by visual inspection and data coinciding with these artifacts were later discarded from further analysis.” (Lines 897 to 901)

The authors write: "We noted that both methods tended to overestimate peak bandwidths (Figure 2 —figure supplement 1)" It seems that there is a systematic bias in peak bandwidth estimation. Is there a way to compensate for it?

The Reviewer is correct in stating that “there is a systematic bias in peak bandwidth estimation”.

This bias is related to the balance of time and frequency resolution inherent to SPRiNT, as well as its consequences on spectral SNR. Donoghue et al., (2020) showed that peak bandwidth estimation error is low when spectral noise is negligible, as is more common for static periodograms (see Figure 3C of Donoghue et al., 2020). Resolving spectral power estimates in time introduces additional noise to the spectrum, which challenges the estimation of spectral parameters (especially peak bandwidth, as we have observed).

We have pasted the relevant portions of the revised manuscript discussing this limitation below:

“The algorithm tends to overestimate the bandwidth of spectral peaks, which we discuss as related to the frequency resolution of the spectrogram (mostly 1 Hz in the present study). The frequency resolution of the spectrogram at 1Hz, for example, may be too low to quantify narrower band-limited components. The intrinsic noise level present in short-time Fourier transforms (i.e., spectral power not explained by periodic or aperiodic components) may also challenge bandwidth estimation. Increasing STFT window length augments spectral resolution and reduces intrinsic noise, although to the detriment of temporal specificity.” (Lines 467 to 474)

It is possible that the changes between young and old participants (or in eyes-open and eyes-closed conditions) were due to relatively local changes in low-frequency oscillations which would consequently lead to deviation from 1/f decay of spectrum. This in turn would lead to changes in goodness-of-fit (GOF) of 1/f component between conditions. Have authors observed systematic changes in GOFs between different conditions?

We appreciate the Reviewer’s insight and suggestion on the interpretation of this aspect of the reported data.

We found that SPRiNT average model fit errors were significantly different between conditions (p = 0.002; two-sample t-test), but not between age groups. However, including model fit error in as a predictor in our logistic regression model for predicting condition did not change the interpretation of previous results. We have now added a table (Table 3) and section to the revised Supplemental Materials to report and detail these findings, provided below:

“We performed t-tests of model fit errors (MAE) between conditions and age groups. While there were no age-related effects on model fit error (eyes-open: p = 0.09; eyes closed: p = 0.69), we observed slightly lower model fit errors in the eyes-open condition (mean = 0.032) compared to the eyes-closed condition (mean = 0.033; t(354) = 3.17, p = 0.002, 95% CI [3.0x10-4 1.3x10-3]). The size of this effect was small-to-medium (Cohen’s d = 0.34).

To determine whether model fit error would affect our SPRiNT logistic regression model for condition, we included it as a fixed effect in a new logistic regression model (Table 3). Here, we observed the same effects for predicting condition as the original model (mean aperiodic exponent, mean α power, variability of α power; see Table 2), with no significant effect of model fit error (p = 0.45).” (Lines 1338 to 1347)

Please discuss cases where the aperiodic part can be stable vs when it can be unstable.

We foresee that new time-resolved methods such as SPRiNT will enable measures of how aperiodic spectrogram parameters evolve over time with behaviour, as illustrated with some of the data in the manuscript where we observe changes in mean aperiodic exponent associated with locomotive behaviour (Figure 5). Aperiodic activity could also exhibit changes in its variability without changing its average shape, though we did not observe this effect in the present study. We have revised the Discussion section by elaborating on a possible scenario (e.g., temporal instability of the excitation/inhibition ratio) where one might expect to observe increased variability in aperiodic parameters, as pasted below:

“Gao et al., (2017) established a link between aperiodic exponent and the local balance of neural excitation vs. inhibition. How this balance adjusts dynamically, potentially over a multiplicity of time scales, and relates directly or indirectly to individual behaviour, demographics, and neurophysiological factors remains to be studied.” (Lines 563 to 567)

Reviewer #2 (Recommendations for the authors):1. In the first analysis, the authors compare SPRiNT with specparam applied to a wavelet time-frequency spectrum. Given that the original method by Donoghue et al., is based on Welch's method (which uses the Discrete Fourier Transform; DFT), it is unclear why the authors chose wavelets as a benchmark. A more direct comparison with Donoghue et al., would be comparing the time-resolved with the static specparam approach, without the implicit comparison of STFT with wavelets. Could the authors please motivate their choice of benchmark? (potentially the analysis described in lines 36-46 of the supplement could be moved/referred to in the main text).

We thank the Reviewer for this insight: we agree that the motivation behind our choice of benchmark was not sufficiently clear in the original manuscript.

Our motivation was to benchmark SPRiNT against alternative approaches to time-resolved parameterization of spectrograms. We used Morlet wavelets to derive comparative spectrograms because these wavelets are extensively used and expected to outperform short-time Fourier transforms in a wide range of signal processing situations (Mallat, 1998).

We agree with the Reviewer that the benchmark comparison of SPRiNT to the static specparam approach is an original aspect that needed to be better emphasized. Therefore, we have moved the specparam results for simulation challenge I to the revised main text, as pasted below:

“We also parameterized the periodogram of each time series of the first simulation challenge with specparam, to assess the outcome of a biased assumption of stationary spectral contents across time. The PSDs were computed using the Welch approach over 1-s time windows with 50% overlap. The average recovered aperiodic exponent was 1.94 Hz-1 (actual = 1.5 to 2 Hz-1) and offset was -1.64 a.u. (actual = -2.56 to -1.41 a.u.). The only peak detected by specparam (99% sensitivity) was the α peak, with an average center frequency of 8.09 Hz (actual = 8 Hz), amplitude of 0.79 a.u. actual max = 1.2 a.u., and peak frequency standard deviation of 1.21 Hz (actual = 1.2 Hz). No β peaks were detected across all spectra processed with specparam.” (Lines 208 to 215)

2. One issue when applying SPRiNT to task data is that it temporally smooths a/periodic parameter estimates (i.e. by averaging Fourier coefficients over neighbouring windows), which can lead to blurring of baseline and task windows (especially in a typical task where baseline and task period are only a few seconds). Could the authors elaborate on how to choose parameter settings (e.g. number of overlapping time windows; whether or not to fit a knee) and what pitfalls to look out for?

The Reviewer's point is well taken.

We now develop these important practical considerations in the Discussion, and provide recommendations to readers and future users of SPRiNT, depending on their neuroscience questions and study objectives. The relevant section is pasted below:

“SPRiNT returns goodness-of-fit metrics for all spectrogram parameters. However, these metrics cannot account entirely for possible misrepresentations or omissions of certain components of the spectrogram. Visual inspections of original spectrograms and SPRiNT parameterizations are recommended e.g., to avoid fitting a ‘fixed’ aperiodic model to data with a clear spectral knee, or to ensure that the minimum peak height parameter is adjusted to the peak of lowest amplitude in the data. Most of the results presented here were obtained with similar SPRiNT parameter settings. Below are practical recommendations for SPRiNT parameter settings, in mirror and complement of those provided by Ostlund et al., (2022) and Gerster et al., (2022) for specparam:

Window length determines the frequency and temporal resolution of the spectrogram. This parameter needs to be adjusted to the expected timescale of the effects under study so that multiple overlapping SPRiNT time windows (with overlap) cover the expected duration of the effect of interest; see for instance, the 2-s time windows with 75% overlap designed to detect the effect at the timescale characterized in Figure 5.

Window overlap ratio is a companion parameter of window length that also determines the temporal resolution of the spectrogram. While a greater overlap ratio increases the rate of temporal sampling, it also increases the redundancy of the data information collected within each time window and therefore smooths the spectrogram estimates over the time dimension. A general recommendation is that longer time windows (>2 s) enable larger overlap ratios (>75%). We recommend a default setting of 50% as a baseline for data exploration.

Number of windows averaged in each time bin enables to control the signal-to-noise ratio (SNR) of the spectrogram estimates (higher SNR with more windows averaged), with the companion effect of increasing the temporal smoothing (i.e., decreased temporal resolution) of the spectrogram. We recommend a baseline setting of 5 windows.

Learning from the specparam experience, we expect that more practical (and critical) recommendations will emerge and be shared by more users adopting SPRiNT, with the pivotal expectation, as with all analytical methods in neuroscience (Salmelin and Baillet, 2009), that users carefully and critically review the sensibility of the outcome of SPRiNT parameterization applied to their own data and to their own neuroscience questions (Ostlund et al., 2022).” Lines (570 to 598)

3. Line 194 refers to a supplemental figure (the figure 2 equivalent prior to removal of outlier peaks). Line 21 of the supplement also refers to this figure. However, the figure appears to be absent.

Thank you for catching this mistake: the figure was indeed erroneously omitted from the original manuscript. The figure is now included as Figure 2 —figure supplement 3.

4. Figure 3D shows there is a general underestimation of the number of periodic components, especially in the δ band. Perhaps it would be useful to add a figure to the supplement containing a confusion matrix, i.e. showing how likely each simulated peak is to be recognised as a peak in a different frequency band (or not recognised) over all simulations.

The suggestion from the Reviewer is very appreciated.

We have produced a confusion matrix for the revised manuscript, which is now displayed in Figure 3 —figure supplement 1b.

Please note that, as we aimed to associate detected peak components with the original periodic components of the synthetized data, the estimated peaks could not be located more than 2.5 peak standard deviations from the simulated centre frequency, which explains why, for instance, no 3-8 Hz peaks were recovered in the 18-35 Hz range.

5. Figure 3C shows the detection probability of spectral peaks with respect to centre frequency and peak amplitude. Could to authors create a similar figure for bandwidth, or at least comment on this?

Thank you for this suggestion.

We have produced and added the proposed figure as Figure 2 —figure supplement 1a in the revised manuscript. We also discuss that the probability of peak detection does not depend on the bandwidth of the periodic component on line 236, pasted below:

“However, the detection rate did not depend on peak bandwidth (Figure 3 —figure supplement 1a).” (Line 235)

6. Alternatives to the Short Time Fourier Transform (STFT) are discussed in both the introduction and discussion but do not mention Empirical Mode Decomposition (EMD; Huang et al., 1998; Quinn et al., 2021).

Thank you for this suggestion.

We now mention EMD in the Introduction; please see lines 53 to 57, pasted below:

“Current methods for measuring the time-varying properties of neural fluctuations include several time-frequency decomposition techniques such as Hilbert, wavelet, and short-time Fourier signal transforms (Bruns, 2004; Cohen, 2014), and more recently, empirical mode decompositions (EMD; Huang et al., 1998) and time-delay embedded hidden Markov models (TDE-HMM; Quinn et al., 2018).” (Lines 53 to 57)

7. Recently, another adaptation of specparam, called PAPTO, was described by Brady and Bardouille (2022; https://doi.org/10.1016/j.neuroimage.2022.118974), specifically regarding transient oscillations. It would be good if the authors could add this in their discussion, especially in light of pruning the periodic component outliers.

Thank you for mentioning this recent published work.

We agree that PAPTO is a relevant approach for detecting transient oscillatory activity (and aperiodic compensation) in the field.

We now discuss the relevance of PAPTO and another methodologically similar approach (fBOSC; Seymour et al., 2022) in the revised Discussion:

“Scientific interest towards aperiodic neurophysiological activity has recently intensified, especially in the context of methodological developments for the detection of transient oscillatory activity in electrophysiology (Brady and Bardouille, 2022; Seymour et al., 2022). These methods first remove the aperiodic component from power spectra using specparam, before detecting oscillatory bursts from wavelet spectrograms. SPRiNT’s outlier peak removal procedure also detects burst-like spectrographic components, although for a different purpose. SPRiNT is one methodological response for measuring and correcting for aperiodic spectral components and as such, could contribute to improve tools for detecting oscillatory bursts, as suggested by Seymour et al., (2022).” (Lines 616 to 624)

8. A few observations from the Results are missing in the discussion. I would like to ask the authors to add a discussion on (1) the overestimation of peak bandwidth by SPRiNT, (2) the underestimation of the number of detected peaks in figure 3, and (3) the peaks in the aperiodic component and offset at moments of switching eyes open/closed in figure 4 – supplement 1 (also related to point 2).

Thank you for pointing at these gaps in the Discussion.

We have revised the Discussion accordingly (with the third point being more methodological), addressing all the points raised by this Reviewer. Please see the following sections of the manuscript, pasted below:

“The algorithm tends to overestimate the bandwidth of spectral peaks, which we discuss as related to the frequency resolution of the spectrogram (mostly 1 Hz in the present study). The frequency resolution of the spectrogram at 1Hz, for example, may be too low to quantify narrower band-limited components. The intrinsic noise level present in short-time Fourier transforms (i.e., spectral power not explained by periodic or aperiodic components) may also challenge bandwidth estimation. Increasing STFT window length augments spectral resolution and reduces intrinsic noise, although to the detriment of temporal specificity.” (Lines 467 to 474)

“We also found that SPRiNT may underestimate the number of periodic components, though this can be interpreted as the joint probability of SPRiNT detecting multiple independent oscillatory peaks (where the probability of detecting a given peak is between 65-75%; approximating a binomial distribution).” (Lines 474 to 474)

“We observed sharp changes in aperiodic exponent and offset at the transitions between eyes-open and eyes-closed (Figure 4 —figure supplement 1), which are likely to be artifactual residuals of eye movements. We discarded these segments from further analysis.” (Lines 296 to 298)

9. From lines 245-248 it does not become clear from the main text that the authors conducted a logistic regression analysis; this only becomes clear from the subtext of figure 4B. Please add it to the main text.

Thank you for highlighting this aspect.

We now explicitly mention that we conducted a logistic regression analysis in the relevant sections of the Results, with variations of the following pasted section:

“We ran a logistic regression model with SPRiNT parameter estimates…” (Lines 298 to 299)

10. It seems that tables 5 and 6 are mixed up in the main text (line 304, 309), e.g. line 304 is referring to table 5 but should be referring to table 6.

Indeed, we are grateful that this Reviewer has identified this table numbering error.

We have now revised the tables concerning the LEMON dataset to include Bayes factors, which also decreases the number of figure supplements. Please also note that we have updated our table numbering such that the referenced tables are now Table 11 for eyes-open and Table 12 for eyes-closed.

11. There appears to be a grammatical error in lines 416-418 ("…has associated how locomotor behavior is associated…").

We have reworded these lines for clarity, thank you.

12. Figure 3D would be clearer with a legend. Also, the subtext talks about "blue" for 3-8 Hz peaks but is ambiguous because (dark) blue also denotes an undetected peak. Please clarify in the text.

Thank you for helping us improve the visual quality and consistency of the figures.

We have reorganized the use of colours in all figures and clarified their correspondences between figures.

13. The reference to figure 3E in the subtext should be in bold font.

We have fixed this oversight; thank you for your attention.